# Decoding Intelligence: A Framework for Certifying Knowledge Comprehension in LLMs

## Abstract

Knowledge comprehension capability is an important aspect of human intelligence. As Large Language Models (LLMs) are being envisioned as superhuman agents, it is crucial for them to be proficient at knowledge comprehension. However, existing benchmarking studies do not provide consistent, generalizable, and formal guarantees on the knowledge comprehension capabilities of LLMs. In this work, we propose the first framework to certify knowledge comprehension in LLMs with formal probabilistic guarantees. Our certificates are quantitative — they consist of high-confidence, tight bounds on the probability that a target LLM gives the correct answer on any knowledge comprehension prompt sampled from a distribution. We design and certify novel specifications that precisely represent distributions of knowledge comprehension prompts leveraging knowledge graphs. We certify SOTA LLMs for specifications over the Wikidata5m knowledge graph. We find that knowledge comprehension improves with increasing model size.

## 1 Introduction

Large Language Models (LLMs) have demonstrated human-level performance for several real-world tasks (Street et al., 2024; Yang et al., 2023b; Bommasani et al., 2022; Harrison, 2024). An important use case for LLMs is knowledge comprehension (Lazaridou et al., 2022; Khattab et al., 2023), that is, they are often used to summarize long texts and webpages (Perplexity, 2023; Nakano et al., 2022), respond to user queries based on the context (Yang et al., 2018), and serve as adaptive task decomposers for reasoning-based retrieval augmented generation tasks (Yao et al., 2023b). Knowledge comprehension involves answering questions by extracting relevant information from large, unstructured texts and reasoning with it. Large context windows of millions of tokens in models like Gemini v1.5 (Gemini Team, 2024) reduce reliance on large knowledge corpora of RAG systems and parametric knowledge held by LLMs. Users increasingly provide extensive references during inference to guide responses. This makes analyzing the knowledge-comprehension and reasoning capabilities of popular LLMs crucial. Moreover, knowledge comprehension is considered a basic evaluation of language understanding in human learners, according to the Bloom's taxonomy (Bloom, 1956). Students are tested on knowledge comprehension tasks at all levels of school education (National Center for Education Statistics, 2024). Standardized tests such as TOEFL (Educational Testing Service, 2024) and IELTS (IDP IELTS, 2024) contain knowledge comprehension as entire assessments. As LLMs are envisioned to become superhuman agents (Xi et al., 2023), it is imperative to formally analyze them on tasks on which humans are extensively tested, like knowledge comprehension.

There are several benchmarking studies on the performance of LLMs for knowledge comprehension (Liang et al., 2023; Chen et al., 2021; Yang et al., 2018; Wang et al., 2023a; Trivedi et al., 2022; Tang & Yang, 2024). Several of these studies use multi-hop question-answering datasets that consist of questions requiring several sequential reasoning steps to obtain the final correct answer. Thus, benchmarking knowledge comprehension often involves analyzing whether the target LLM can combine multiple pieces of information in meaningful ways and reason its way to the correct answer in the prompt, without deviating or hallucinating. However, the empirical nature of prior studies results in inconsistency in their observations (Wei et al., 2023b; Olsson et al., 2022; Shi et al., 2024). Moreover, while they can convey some high-level trends in the performance of popular LLMs, the results are devoid of any formal guarantees on their applicability. Such guarantees are crucial when deploying LLMs in large-scale knowledge-comprehension tasks in critical domains such as medicine or finance, as they give more confidence about the trustworthiness of LLMs before

deployment. Our work aims to bridge this gap by introducing a novel formal certification method, QuaCer-C[1], for obtaining certified bounds on the knowledge comprehension capability of LLMs.

**Key challenges**. We face the following challenges when developing a formal certification framework for knowledge comprehension in LLMs. (1) We need formal representations capturing the knowledge comprehension property, amenable to certification. Such representations should precisely capture large, diverse sets of prompts (created by varying questions, supporting texts, etc.) for knowledge comprehension and their correct responses. (2) Failure examples where the desirable property does not hold are fairly easy to construct for LLMs by appropriate prompt tuning (Xu et al., 2024; Vega et al., 2023), making binary certificates that indicate whether an LLM satisfies a specification trivially false. (3) Giving provable guarantees on LLMs is a hard, open problem, due to the high number of model parameters and nonlinearities (Zhang et al., 2024), for which the traditional certifiers (Singh et al., 2019; Shi et al., 2020; Bonaert et al., 2021) would lose significant precision leading to inconclusive analysis. The number of prompts over which we desire the target LLMs to reason correctly is also large, making enumeration-based methods for formal guarantees infeasible.

**Our approach**. We formalize knowledge comprehension as a novel specification using knowledge graphs. Instead of specifying correctness of LLM responses for all prompts in any given set, we specify a quantitative property, which is the probability of correct response for any knowledge comprehension prompt sampled from a distribution, developed using a given knowledge graph. We propose a black-box quantitative certification approach, QuaCer-C, which circumvents the issues that traditional approaches have with the number of parameters in LLMs and can even work for closed-source API-access LLMs. QuaCer-C generates high-confidence bounds on the quantitative property using queries, leveraging binomial proportion confidence intervals (Clopper & Pearson, 1934). Estimating with bounds is beneficial as they also account for the uncertainty in the estimation. While formal analysis has been conducted on individual generations of LLMs in prior work (Quach et al., 2024), there is no analysis for the average-case risk of LLMs in knowledge comprehension. Our certificates contain provable bounds on the probability of getting correct responses for any random knowledge comprehension task sampled from the distributions given in the specifications.

**Contributions**. We make the following contributions:

1. We specify the knowledge comprehension property desirable from the LLM responses as a formal specification. Our specifications use popular knowledge graphs such as Wikidata5m (Wang et al., 2021) that are augmented with supporting information about each of their entities. The specifications represent a large set of knowledge comprehension prompts with their respective correct answers expected from any target LLM.

2. We model certification in a target LLM as a probability estimation problem and leverage Clopper-Pearson confidence intervals to generate provable, high-confidence bounds on the quantitative property of interest. Our implementation is provided at `https://anonymous.4open.science/r/QuaCer_CAnon-4130`.

3. We generate the proposed certificates for the popular LLMs: Llama-3, Mistral, Phi-3, GPT-4o, and Gemini-1.5-Pro. We observe that as the number of model parameters increases, the knowledge comprehension capability of the LLM improves. On comparing different model classes, we see Phi-3 models performing the best among the smaller, open-source models.

Our work is the first step towards providing guarantees on the knowledge comprehension capabilities of LLMs, to ameliorate the caution needed when using LLMs (Shanahan, 2023) in a systematic way. We anticipate it to go a long way in making LLMs trustworthy for deployment in critical domains.

## 2 BACKGROUND

### 2.1 KNOWLEDGE GRAPH

A knowledge graph $\mathcal{G} = (\mathcal{N}, \mathcal{E})$ is a collection of nodes $\mathcal{N}$ representing entities, interconnected by directed edges $\mathcal{E}$ representing their relations (Peng et al., 2023; Ji et al., 2022). They are commonly used in search engines to enhance the relevance of responses to user queries.

---

[1]**Qua**ntitative **Cer**tification of Knowledge **C**omprehension

Hence, major companies develop their own closed-source knowledge graphs. Wikidata5m (Wang et al., 2021), a popular open-source knowledge graph consisting of 5M nodes, is a structured representation of Wikipedia pages. Each Wikidata node corresponds to a Wikipedia page, containing its abstract and a set of aliases that can synonymously refer to the node. Two nodes $(v_1, v_2)$, $v_1, v_2 \in \mathcal{N}$ are connected by a labeled edge if there is a link in the supporting document for $v_1$ to that for $v_2$. Edge $(v_1, v_2)$ is labeled by a set of aliases for the relation between $v_1$ and $v_2$. Figure 1 shows a subgraph of Wikidata5m.

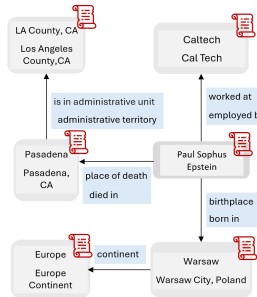

Figure 1: A subgraph of Wikidata5m

## 2.2 Large Language Models

Large Language Models (LLMs) are autoregressive causal language models that operate on a vocabulary $\mathcal{V}$, a set of tokens. LLMs takes a sequence of tokens $x_1, ..., x_n$ where $x_i \in \mathcal{V}, n > 0$ and outputs a probability distribution over $\mathcal{V}$ for the potential next token $x_{n+1}$. These models are typically pretrained on vast corpora of text data (Liu et al., 2024) and have shown remarkable capabilities (Touvron et al., 2023; Gemini Team, 2024; OpenAI, 2024). Numerous benchmarks (Yang et al., 2018; Rein et al., 2023; Hendrycks et al., 2021) have been developed to evaluate the performance of LLMs on tasks related to multi-step reasoning, knowledge comprehension and question answering. However, there remains a gap in our theoretical understanding of LLMs' capabilities.

## 2.3 Information extraction and reasoning

Information extraction (IE) and reasoning are important research problems in natural language processing. IE involves "extracting structured information from unstructured or semi-structured data" (Chen et al., 2022) such as textual documents. Examples of IE are event extraction (Wadden et al., 2019) and relationship extraction (Pawar et al., 2017). Reasoning is the ability of a model to connect multiple facts using correct logical operations to arrive at a final answer (Huang & Chang, 2023). Typically, reasoning capabilities of LLMs are enhanced by using techniques such as Chain of Thought reasoning and its variants (Wei et al., 2023a; Yao et al., 2023a; Wang et al., 2023b), using world models (Hao et al., 2023), etc. It is evaluated in several tasks such as planning (Wang et al., 2024), mathematical reasoning (Imani et al., 2023), commonsense reasoning (Zhao et al., 2023), etc.

## 3 Certifying knowledge comprehension

Knowledge comprehension is the ability of a model to accurately reason through a multi-hop question (Yang et al., 2018) (combination of multiple simple information extraction questions that should be answered sequentially to arrive at the final answer) and extract the answers to intermediate questions from information provided in its prompt to reach the correct final answer. Thus, knowledge comprehension is a combination of reasoning and information extraction. Figure 2 gives an overview of our certification framework, QuaCer-C. We formally define knowledge comprehension using a knowledge graph $\mathcal{G} = (\mathcal{N}, \mathcal{E})$ (Section 2.1) next. Our framework is agnostic to the internal structure of the target model $\mathcal{L}$ which can be any text-to-text generating model.

### 3.1 Specifying knowledge comprehension

The adjacent grammar defines a knowledge graph. Let $\mathcal{V}$ denote the vocabulary of $\mathcal{L}$'s tokens. $\mathcal{V}^+$ denotes the set of concatenation of non-empty sequences of elements of $\mathcal{V}$. Let each node $v$ of $\mathcal{G}$ (line 4) consist of a finite list of synonymous names (a.k.a. aliases, $\mathcal{A}$) that can be used to refer to the node, and a context $\gamma$ that provides more information about the node and its relations with other nodes. For example, in Wikidata5m (Wang et al., 2021), each node has aliases consisting of the identifiers mentioned for the subject of the corresponding Wikipedia page and context which is the abstract of the page. Each edge $e$ in $\mathcal{G}$ (line 5) is an ordered pair of related

| Knowledge Graph Grammar |
| :--- |

| | | |
| :--- | :--- | :--- |
| 1. $\gamma$ | := | $\mathcal{V}^+$ |
| 2. $\eta$ | := | $\mathcal{V}^+$ |
| 3. $\mathcal{A}$ | := | $[\eta_1, \eta_2, \dots]$ |
| 4. $v$ | := | $(\gamma, \mathcal{A})$ |
| 5. $e$ | := | $((v_1, v_2), \mathcal{A})$ |
| 6. $\mathcal{N}$ | := | $[v_1, v_2, \dots]$ |
| 7. $\mathcal{E}$ | := | $[e_1, e_2, \dots]$ |
| 8. $\mathcal{G}$ | := | $(\mathcal{N}, \mathcal{E})$ |

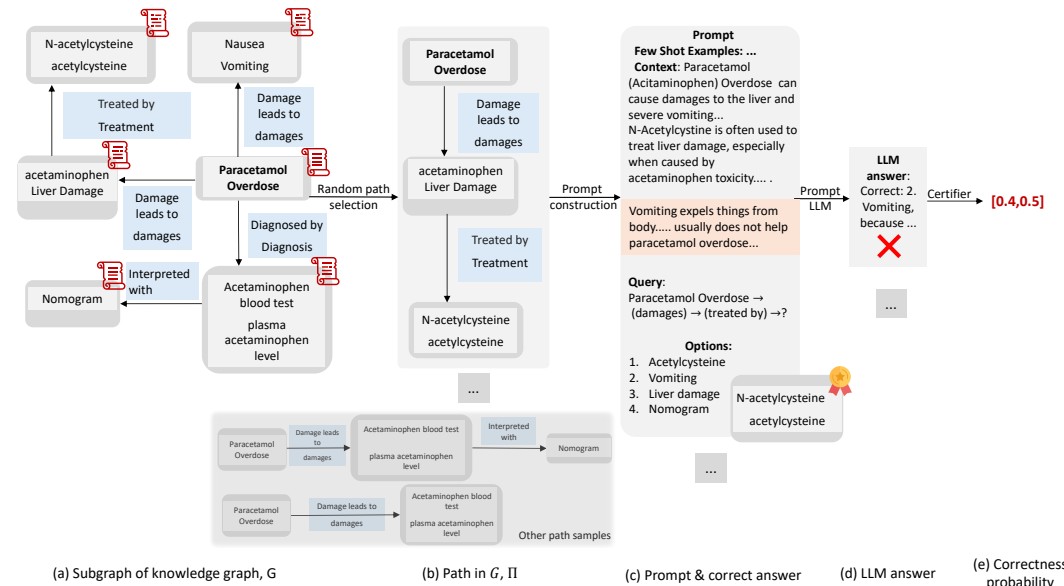

Figure 2: Overview of QuaCer-C. (a) A knowledge graph $\mathcal{G}$ pivoted on some node $v_1$, in this case on 'Paracetamol Overdose'. (b) A randomly chosen path $\Pi$ originating at $v_1$ from the various other possible paths from $v_1$ in $\mathcal{G}$. (c) A prompt created from $\Pi$ having contexts of the nodes in $\Pi$, a distractor context (highlighted in orange, as the node for 'Vomiting' is a distractor for $\Pi$), and a query from $\Pi$. (d) The target LLM's response to the prompt, validated using the correct answer. (e) Certifier obtains bounds on the probability of correct response using $n$ samples of LLM responses.

nodes where the relation is identified by a set of synonymous aliases $\mathcal{A}$ for the edge. Let $(v_1, v_2)$ denote any edge between nodes $v_1$ and $v_2$ in $\mathcal{G}$. We define $\mathcal{G}$ (line 8) as a finite collection of nodes $\mathcal{N}$ and edges $\mathcal{E}$. A path in $\mathcal{G}$ (Definition 3.1) is a set of connected nodes in $\mathcal{N}$.

**Definition 3.1.** (Path in a Knowledge Graph). A path $\Pi = [v_1, v_2 \ldots, v_l]$ is an ordered collection of nodes in a given knowledge graph $\mathcal{G}$, where $l > 1$, such that $\forall i \in [l-1], v_i \in \mathcal{N}, (v_i, v_{i+1}) \in \mathcal{E}$, and $\forall j \in [l], i \neq j \implies v_i \neq v_j$. $\Pi_H := v_1$ and $\Pi_T := v_l$ are the *head* and *tail* nodes respectively of $\Pi$. Let the $i^{th}$ ($i \in [1, l]$) nodes of $\Pi$ from $\Pi_H$ and backwards from $\Pi_T$ be $\Pi[i] := v_i$ and $\Pi[-i] := v_{l-i+1}$ respectively.

Definition 3.2 describes a multi-hop reasoning problem, derived from a given knowledge graph $\mathcal{G}$. As $\mathcal{G}$ naturally encodes several multi-hop problems, we use it to form the specification for a target language model $\mathcal{L}$, similar to prior works such as (Ho et al., 2020; Jiang et al., 2023b).

**Definition 3.2.** (Multi-hop reasoning problem from $\mathcal{G}$). Consider any path $\Pi$ (Definition 3.1) of length $l$ in $\mathcal{G}$. A multi-hop reasoning problem $\mathcal{Q}$ for $\Pi$ is identifying the tail node $\Pi_T$, given an alias of its head node $\Pi_H$ and aliases of all edges from $\Pi_H$ to $\Pi_T$ in $\Pi$. Let $v_{\mathcal{A}}$ denote the corresponding aliases of node $v$ in $\mathcal{G}$. Let $\mathcal{D}$ be a function that samples a random alias from the given set of aliases.

$$\mathcal{Q} := \mathcal{D}(\Pi_{H,\mathcal{A}}) \xrightarrow{\mathcal{D}((\Pi_H, \Pi[2])_{\mathcal{A}})} \ldots \xrightarrow{\mathcal{D}((\Pi[-2], \Pi_T)_{\mathcal{A}})}?$$

$\mathcal{Q}$ thus involves $l - 1$ reasoning steps to get to the final answer, where each step requires correctly identifying intermediate nodes of $\Pi$. Note, however, that correctness of intermediate reasoning steps is generally not evaluated, and accuracy is defined for the final response (Rajpurkar et al., 2016).

To aid $\mathcal{L}$ in correctly answering a multi-hop reasoning query $\mathcal{Q}$ and reduce hallucination (Dhuliawala et al., 2023), we provide relevant textual information needed to identify the intermediate and final nodes in the prompt. Hence, the overall task of answering $\mathcal{Q}$ involves information extraction for identifying intermediate nodes and reasoning to connect the intermediate answers to reach the final answer, which we collectively call *knowledge comprehension*. Our overall property quantifies the probability of observing correct knowledge comprehension for a random multi-hop reasoning query

---

**Algorithm 1** Knowledge comprehension specification

---

**Input:** $\mathcal{L}, \mathcal{G}, v_1, \rho$
**Output:** $p$

1: $\mathcal{D} \coloneqq \mathcal{U} \mid Ber \mid \ldots$
2: $\Pi \coloneqq [v_1, v_2 \coloneqq (\mathcal{D}([v' \mid (v_1, v') \in \mathcal{E}])), \ldots, v_{\mathcal{D}(\{2, \ldots, \rho\})} \coloneqq (\mathcal{D}([v' \mid (v_k, v') \in \mathcal{E}]))]$
3: $\mathcal{Q} \coloneqq \mathcal{D}(\Pi[0]_{\mathcal{A}}) \xrightarrow{\mathcal{D}(\Pi[0], \Pi[1])_{\mathcal{A}}} \ldots \xrightarrow{\mathcal{D}(\Pi[-2], \Pi[-1])_{\mathcal{A}}} \text{\bf ?}$
4: $\Gamma \coloneqq \texttt{shuffle}([\Pi[0]_{\gamma}, \ldots, \Pi[-1]_{\gamma}, (\mathcal{D}(\mathcal{N}))_{\gamma}, \ldots])$
5: $\mathcal{P} \coloneqq \Gamma \odot \mathcal{Q}$
6: $p \coloneqq \texttt{estimateProbability}(\texttt{any}(\mathcal{L}(\mathcal{P}) == \Pi[-1]_{\mathcal{A}}))$

---

developed from $\mathcal{G}$. We formally define the property for $\mathcal{L}$ as a probabilistic program over $\mathcal{G}$ in Algorithm 1. We follow the syntax of the imperative probabilistic programming language in (Sankaranarayanan et al., 2013, Figure 3). The language has primitives for sampling from common distributions such as Uniform ($\mathcal{U}$), Bernoulli ($Ber$), etc., and an $\texttt{estimateProbability(.)}$ function that outputs the probability of a random variable attaining a certain value. As all the random sampling steps in Algorithm 1 can operate with any discrete distribution, we use a generic identifier $\mathcal{D}$ (line 1) for samplers of discrete distributions ('...' denotes samplers for other discrete distributions). We use a primitive function $\texttt{any(.)}$ to denote that at least 1 of its inputs evaluates to true.

As a real-world $\mathcal{G}$ like Wang et al. (2021) can consist of millions of nodes, specifications on the full $\mathcal{G}$ would be impractical as it is hard to certify global specifications over large input spaces (Katz et al., 2017; Geng et al., 2023). Hence, we scope our analyses to local specifications, defined on a subgraph of $\mathcal{G}$ centered on a randomly selected *pivot* node $v_1$ and consisting of all paths originating from $v_1$. Let $\Pi$ be a path in $\mathcal{G}$ that has a randomly selected length $l \in \{2, \ldots, \rho\}$, formed by random sampling of connecting nodes, as described in line 2. From a practical standpoint, queries on longer paths can become meaningless (e.g., Paul Sophus Epstein $\xrightarrow{\text{place of death}} \xrightarrow{\text{administrative unit}} \xrightarrow{\text{country}} \xrightarrow{\text{popular artist}} \xrightarrow{\text{genre}}$ **?**), and thus shorter path lengths are considered in popular multi-hop question-answering datasets such as (Yang et al., 2018; Trivedi et al., 2022). Thus, we upper-bound the lengths of paths (number of nodes in the path) considered in the specification, by a hyperparameter $\rho$. We form a multi-hop reasoning query $\mathcal{Q}$ from $\Pi$ in line 3. The pivot node $v_1$ and the relations are represented by their randomly sampled aliases in $\mathcal{Q}$.

A prompt for $\mathcal{L}$ consists of $\mathcal{Q}$ and a context $\Gamma$ containing information relevant to answer $\mathcal{Q}$. $\Gamma$ is formed by concatenating ($\odot$) the contexts for all nodes in $\Pi$. Let $v_{\gamma}$ denote the corresponding context of node $v$ in $\mathcal{G}$. Prior works (Shi et al., 2023) on analyzing reasoning in LMs have shown the negative influence of irrelevant information (*distractor*) in prompts on the performance of LMs, which is not ideally expected. Hence, we include distractor texts in $\Gamma$ and specify that the correct response should not be based on the distractor information. The contexts of nodes $\tilde{v}$ adjacent to any node $\Pi[i]$ ($i \in [1, l-2]$) on $\Pi$, such that the relation of $(\Pi[i], \tilde{v})$ is the same as that of $(\Pi[i], \Pi[i+1])$, can serve as effective distractors for $\mathcal{Q}$ (Definition 3.3). This is because, at any intermediate step, the model can pick $\tilde{v}$ as the response, which can deviate $\mathcal{L}$'s reasoning from $\Pi$. Nodes adjacent to $\Pi[-1]$ and $\Pi[-2]$ are not distractors. For the former, the model must have already reached the final answer before getting to its adjacent nodes, hence, answering $\mathcal{Q}$. In the latter, adjacent nodes following same relation are valid correct answers and not distractors. We denote distractor text in $\Gamma$ as the context of randomly sampled nodes from a distribution $\mathcal{D}$ over all distractor nodes of $\Pi$ in $\mathcal{N}$. We demonstrate the effects of using distractor text on the performance of SOTA LLMs in Section 4.

**Definition 3.3.** (Distractor node). A distractor node $\tilde{v}$ for a path $\Pi = [v_1, v_2, \ldots, v_l]$ of $\mathcal{G}$ is such that $\forall i \in [1, l], \tilde{v} \neq v_i$, and $\exists j \in [1, l-2], [(v_j, \tilde{v}) \in \mathcal{E}] \wedge [(v_j, \tilde{v})_{\mathcal{A}} = (v_j, v_{j+1})_{\mathcal{A}}]$.

Prior works such as (Chen et al., 2024) have shown that LLM performance can vary with information ordering. Hence, we shuffle the information in $\Gamma$ (line 4) to specify that the model's response should be invariant to the ordering of information. Our final specification in line 6 is the probability that $\mathcal{L}$ generates any alias of the last node of the path, which is the correct answer to $\mathcal{Q}$. The specification depends on the choices for the different distributions used at various sampling steps, $\mathcal{G}, v_1$, and $\rho$. It leads to certificates for the behavior of $\mathcal{L}$ on a given subgraph of $\mathcal{G}$ and paths of length at most $\rho$.

## 3.2 CERTIFICATION METHOD

Our algorithm certifies the target LLM $\mathcal{L}$ by computing an interval $[p_l, p_u]$ containing the value of the probability $p$ (Algorithm 1, line 6) for a given pivot node $v_1$ in $\mathcal{G}$ with high confidence. We model $p$ as the probability of setting the underlying boolean random variable $\mathcal{R} \triangleq (\texttt{any}(\mathcal{L}(\mathcal{P}) == \Pi[-1]_{\mathcal{A}}))$ to true (success). Thus, $\mathcal{R} \sim Ber(p)$. Exactly determining $p$ would require enumerating over all possible $\mathcal{P}$ which can be developed from any path from a subgraph of $\mathcal{G}$ with any random aliases, resulting in an infeasible number of possible prompts, as shown in Appendix B.6. Moreover, we want our method to generalize to closed-source LLMs as well, where the internal structures of the models are unknown. Hence, we cannot use any symbolic methods (Mirman et al., 2020) to determine $p$. Thus, to scalably certify the black-box target LM $\mathcal{L}$, we estimate $p$ with provably high-confidence (low error) bounds. Confidence is defined as the probability of the true value $p$ being within the bounds, i.e., $Pr[p \in [p_l, p_u]]$. To establish formal guarantees, we want our estimation procedure to be such that the actual confidence is at least the user-specified confidence level, $1 - \delta$ (i.e., $Pr[p_l \leq p \leq p_u] \geq 1 - \delta$), where $\delta$ is a small positive constant. Hence we use the conservative method of Clopper-Pearson confidence intervals (Clopper & Pearson, 1934; Brown et al., 2001; Kurz et al., 2014), which is known to produce intervals that are guaranteed to have high confidence. To compute high-confidence bounds on $p$, we make $n$ independent and identically distributed observations of $\mathcal{R}$, in which we obtain $k$ successes, $k \in [0, n]$. We generate Clopper-Pearson confidence intervals with the $n$ observations to bound $p$ with $1 - \delta$ confidence.

## 4 EXPERIMENTS

We certify the following open-source, instruction finetuned (Wei et al., 2022a) models — Llama-3-instruct 8B model (Dubey et al., 2024), Mistral 7B-Instruct-v0.2 (Jiang et al., 2023a), Phi-3 3B and 14B parameter models (Abdin et al., 2024). We also certify 4-bit and 8-bit quantized versions of the open-source models to study the effects of quantization on a model's knowledge comprehension capabilities. Among the closed-source models with API access, we certify Gemini-1.5 (Gemini Team, 2024) Flash-001 and Pro-002 models and GPT-4o-0827 (OpenAI, 2024).

We use Wikidata5m (Wang et al., 2021) as our knowledge graph after preprocessing (check Appendix B.1.1 for details). To generate challenging and diverse specifications, we sample 50 pivot nodes from two populations: the top 2000 nodes by out-degree in the global graph, and nodes whose subgraph within radius $\rho$ contains at least 2000 vertices. This strategy ensures specifications rooted around any of the pivot nodes have a large number of paths, making enumerative certification (where all possibilities are tested for satisfaction of the specification) impractical. Note that QuaCer-C is not limited to such subgraphs, and owing to their challenges in terms of prohibitively large number of possibilities, we select them, only for illustration purposes. We set the maximum path length parameter as $\rho = 5$, as we empirically observe that longer paths could result in queries that are very unrelated to the head node of the path. As our certificates are over all paths with lengths at most $\rho$ in a given subgraph, we equally prioritize the different possible path lengths in $[1, \rho]$, even though paths with longer lengths can be fewer in number than those with shorter lengths. Hence, we define our sampler from our distribution over paths (Algorithm 1, line 2) which first selects a path length from the uniform distribution over the integers $[1, \rho]$. We then sample a path of the chosen length from a uniform distribution over all paths of that length in the subgraph. This ensures that each possible path length, and each path of a given length, has equal probability of selection, preventing bias by prioritizing some elements of the underlying sample spaces. Note, however, the framework is adaptable to other modified distributions as needed by specific certification usecases. (We analyze the impact of varying path lengths on LLM performance in Appendix A.)

Given a path , we can construct a query by uniformly sampling any aliases for the nodes in the path (Algorithm 1, line 3). For instance, for the path with the nodes [Chandler Bing, Matthew Perry, 19 August 2019], a query could be "Chandler Bing→(actor)→(birth date)→?". This query tests the LLM's ability to correctly identify the terminal entity ('?') given the starting entity ('Chandler Bing') and the specified relational path. Additional details and figures provided in Appendix B.2.2.

Following query selection, we construct prompts to evaluate LLM knowledge comprehension. Each prompt includes a query and relevant context, presented as a multiple-choice question. This format allows for straightforward evaluation of LLM responses using string matching (Appendix B.7). Fur-

thermore, we include a fixed set of few-shot examples (Appendix B.5) in the prompt to ensure the LLM understands the task structure. We investigate the impact of varying the number of few-shot examples on LLM performance in Appendix A.

The context accompanying each query depends on kind of specification, which we elaborate on in Section 4.1. Occasionally, we adjust the context to fit within model-specific context window constraints (See appendix B.4 for details). The distribution of generated prompts is primarily determined by the query generation, answer option selection processes, and some context length adjustments. Queries are derived from paths with uniformly distributed lengths, ensuring a balanced representation of reasoning complexity. Answer options are sampled non-uniformly. In our 'distractor setting,' we prioritize distractors, followed by entities from the query path, and finally, randomly selected entities related to nodes in the path. This approach strikes a balance between challenging the LLM with complex queries and presenting diverse, potentially misleading answer choices. A detailed description of the prompt construction process can be found in Appendix B.3. QuaCer-C generates certificates with confidence $1 - \delta = 95\%$ and number of samples, $n = 250$ samples. We conduct experiments for open-source LLMs on 2 A100 GPUs with 40GB VRAM each.

**Baseline**. We compare QuaCer-C's results with a benchmarking baseline. This baseline consists of the accuracy of the target LLM on a static dataset. We make this dataset with 50 randomly chosen paths in each subgraph with which we also form the specifications, certified using QuaCer-C. The queries are formed with the first-occurring aliases of the entities in their corresponding contexts. The prompts are generated as in the vanilla setting from each query.

### 4.1 DIFFERENT KINDS OF SPECIFICATIONS

We study the certificates for 3 different kinds of specifications that arise from variations in the construction of context in the prompts to LLMs (Algorithm 1, line 4) — with context shuffling and distractor context (Shuffle Distractor), with context shuffling and without distractor context (Shuffle), and without context shuffling and without distractor context (Vanilla). These settings enable us to study the effects of these operations on the knowledge provided in LLM prompts. When distractor context is provided, it is only for 1 distractor node, so as to fit the relevant context for the nodes on the considered path within the context windows of the target LLMs. We hypothesize that distractors to nodes later in the path, closer to the tail node which consists of the final answer, would be more challenging for the LLM due to their proximity and similarity to the answer node. Our distractor node sampler (Appendix B.3, Algorithm 4) thus employs a weighted sampling approach to prioritize distractor nodes closer to the path's tail. We provide an ablation study on the effects of varying the distribution of distractor nodes in Appendix A.

### 4.2 CERTIFICATES

QuaCer-C generates certificates providing high-confidence, tight lower and upper bounds on the probability of a correct LLM response to a random prompt sampled using the prompt constructor from the given distribution of prompts in our specifications. We report the average value of the lower and upper bounds, over our set of specifications that QuaCer-C certifies for each LLM, in Table 1. We also report the average empirical probability (the ratio of correct responses to the total number of prompts, $n$, for each certificate), averaged over the test set. We show further certification results with chain-of-thought reasoning (Wei et al., 2023a) in Appendix A.4. Next, we summarize the key observations from the certification results in Table 1, some of which follow trends from prior works.

**Scaling of knowledge comprehension with model size**. We observe that the larger models such as Gemini and GPT have significantly higher bounds than those for the smaller models such as Phi-3, Mistral, Llama (Figure 5). The lower bounds of the larger models are higher than the upper bounds of the smaller models, suggesting a paradigm shift in knowledge comprehension capabilities (especially for Gemini-1.5-Pro and GPT-4o). However, as the larger models are also closed-source, we are unaware whether the enhanced knowledge comprehension

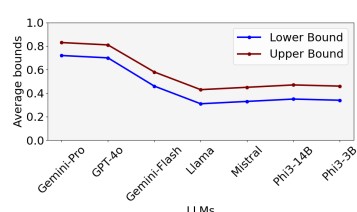

Figure 5: Variation in certification bounds with models (Vanilla, fp16)

Table 1: Certification Results for Different LLMs

| Model | Precision | Baseline | Specification Kind | Avg. Lower Bound | Avg. Upper Bound | Avg. Accuracy |
|---|---|---|---|---|---|---|
| Gemini -1.5-Pro | - - - | $0.83 \pm 0.06$ | Vanilla
Shuffle
Shuffle Distractor | $0.72 \pm 0.06$
$0.71 \pm 0.06$
$0.64 \pm 0.09$ | $0.83 \pm 0.05$
$0.82 \pm 0.06$
$0.75 \pm 0.08$ | $0.78 \pm 0.06$
$0.77 \pm 0.06$
$0.70 \pm 0.09$ |
| GPT-4o | - - - | $0.84 \pm 0.07$ | Vanilla
Shuffle
Shuffle Distractor | $0.70 \pm 0.06$
$0.69 \pm 0.06$
$0.62 \pm 0.09$ | $0.81 \pm 0.06$
$0.80 \pm 0.06$
$0.74 \pm 0.08$ | $0.76 \pm 0.06$
$0.75 \pm 0.06$
$0.68 \pm 0.09$ |
| Gemini -1.5-Flash | - - - | $0.72 \pm 0.08$ | Vanilla
Shuffle
Shuffle Distractor | $0.46 \pm 0.06$
$0.45 \pm 0.06$
$0.42 \pm 0.10$ | $0.58 \pm 0.06$
$0.57 \pm 0.06$
$0.55 \pm 0.10$ | $0.52 \pm 0.06$
$0.51 \pm 0.06$
$0.48 \pm 0.10$ |
| Llama (8B) | fp16 | $0.49 \pm 0.09$ | Vanilla
Shuffle
Shuffle Distractor | $0.31 \pm 0.09$
$0.31 \pm 0.05$
$0.30 \pm 0.10$ | $0.43 \pm 0.10$
$0.44 \pm 0.06$
$0.42 \pm 0.11$ | $0.36 \pm 0.10$
$0.37 \pm 0.06$
$0.36 \pm 0.11$ |
| | 8bit | $0.44 \pm 0.09$ | Vanilla
Shuffle
Shuffle Distractor | $0.30 \pm 0.05$
$0.31 \pm 0.06$
$0.30 \pm 0.06$ | $0.42 \pm 0.06$
$0.44 \pm 0.06$
$0.42 \pm 0.06$ | $0.36 \pm 0.06$
$0.37 \pm 0.06$
$0.35 \pm 0.06$ |
| | 4bit | $0.43 \pm 0.09$ | Vanilla
Shuffle
Shuffle Distractor | $0.27 \pm 0.05$
$0.28 \pm 0.07$
$0.25 \pm 0.09$ | $0.39 \pm 0.05$
$0.40 \pm 0.07$
$0.36 \pm 0.09$ | $0.33 \pm 0.05$
$0.34 \pm 0.07$
$0.30 \pm 0.09$ |
| Mistral (7B) | fp16 | $0.52 \pm 0.08$ | Vanilla
Shuffle
Shuffle Distractor | $0.33 \pm 0.05$
$0.34 \pm 0.05$
$0.33 \pm 0.05$ | $0.45 \pm 0.05$
$0.46 \pm 0.06$
$0.45 \pm 0.05$ | $0.39 \pm 0.05$
$0.40 \pm 0.05$
$0.39 \pm 0.05$ |
| | 8bit | $0.52 \pm 0.09$ | Vanilla
Shuffle
Shuffle Distractor | $0.32 \pm 0.05$
$0.34 \pm 0.06$
$0.33 \pm 0.11$ | $0.44 \pm 0.06$
$0.46 \pm 0.06$
$0.46 \pm 0.12$ | $0.38 \pm 0.05$
$0.40 \pm 0.06$
$0.39 \pm 0.12$ |
| | 4bit | $0.49 \pm 0.08$ | Vanilla
Shuffle
Shuffle Distractor | $0.31 \pm 0.06$
$0.32 \pm 0.05$
$0.28 \pm 0.11$ | $0.43 \pm 0.06$
$0.44 \pm 0.06$
$0.39 \pm 0.12$ | $0.37 \pm 0.06$
$0.38 \pm 0.06$
$0.33 \pm 0.11$ |
| Phi-3 (14B) | fp16 | $0.58 \pm 0.08$ | Vanilla
Shuffle
Shuffle Distractor | $0.35 \pm 0.05$
$0.35 \pm 0.04$
$0.33 \pm 0.11$ | $0.47 \pm 0.05$
$0.48 \pm 0.04$
$0.45 \pm 0.11$ | $0.41 \pm 0.05$
$0.41 \pm 0.04$
$0.38 \pm 0.11$ |
| | 8bit | $0.46 \pm 0.06$ | Vanilla
Shuffle
Shuffle Distractor | $0.35 \pm 0.05$
$0.34 \pm 0.06$
$0.31 \pm 0.08$ | $0.47 \pm 0.05$
$0.47 \pm 0.06$
$0.43 \pm 0.09$ | $0.41 \pm 0.05$
$0.40 \pm 0.06$
$0.37 \pm 0.08$ |
| | 4bit | $0.43 \pm 0.08$ | Vanilla
Shuffle
Shuffle Distractor | $0.33 \pm 0.04$
$0.33 \pm 0.04$
$0.30 \pm 0.08$ | $0.45 \pm 0.05$
$0.46 \pm 0.05$
$0.42 \pm 0.09$ | $0.39 \pm 0.05$
$0.39 \pm 0.05$
$0.36 \pm 0.09$ |
| Phi-3 (3B) | fp16 | $0.50 \pm 0.09$ | Vanilla
Shuffle
Shuffle Distractor | $0.34 \pm 0.05$
$0.34 \pm 0.04$
$0.32 \pm 0.10$ | $0.46 \pm 0.06$
$0.47 \pm 0.05$
$0.45 \pm 0.10$ | $0.40 \pm 0.05$
$0.40 \pm 0.05$
$0.38 \pm 0.10$ |
| | 8bit | $0.44 \pm 0.06$ | Vanilla
Shuffle
Shuffle Distractor | $0.32 \pm 0.05$
$0.32 \pm 0.04$
$0.31 \pm 0.09$ | $0.44 \pm 0.06$
$0.44 \pm 0.05$
$0.43 \pm 0.10$ | $0.38 \pm 0.06$
$0.38 \pm 0.05$
$0.37 \pm 0.10$ |
| | 4bit | $0.43 \pm 0.08$ | Vanilla
Shuffle
Shuffle Distractor | $0.32 \pm 0.05$
$0.32 \pm 0.05$
$0.29 \pm 0.10$ | $0.44 \pm 0.06$
$0.44 \pm 0.05$
$0.41 \pm 0.10$ | $0.38 \pm 0.06$
$0.38 \pm 0.05$
$0.35 \pm 0.10$ |

capabilities could be due to specialized training or finetuning techniques applied on the models. We see that the smaller models have similar certification bounds. Interestingly, Phi3-3B, which is the smallest model we consider, is performing comparatively to its 14B counterparts and to larger Mistral and Llama models. This contradicts works such as (Wei et al., 2022b; Qin et al., 2024) claiming that reasoning capabilities emerge only when model parameters scale to tens or hundreds of billions.

**Effects of model quantization**. We see that higher quantization deteriorates model performance on knowledge comprehension, contrary to prior works like Jin et al. (2024) that suggest that 4-bit quantization can retain the model's knowledge and reasoning capabilities.

**Effects of different kinds of specifications**. Our results for the different kinds of specifications — Vanilla, Shuffle, and Shuffle Distractor, indicate that the Vanilla specifications are generally easier, resulting in higher certification bounds. Shuffle Distractor specifications are challenging specifications for all models resulting in consistently lower certification bounds. However, the differences in the bounds' values are not high across the settings, potentially due to the challenge of identifying relevant information from large and unstructured contexts in all cases.

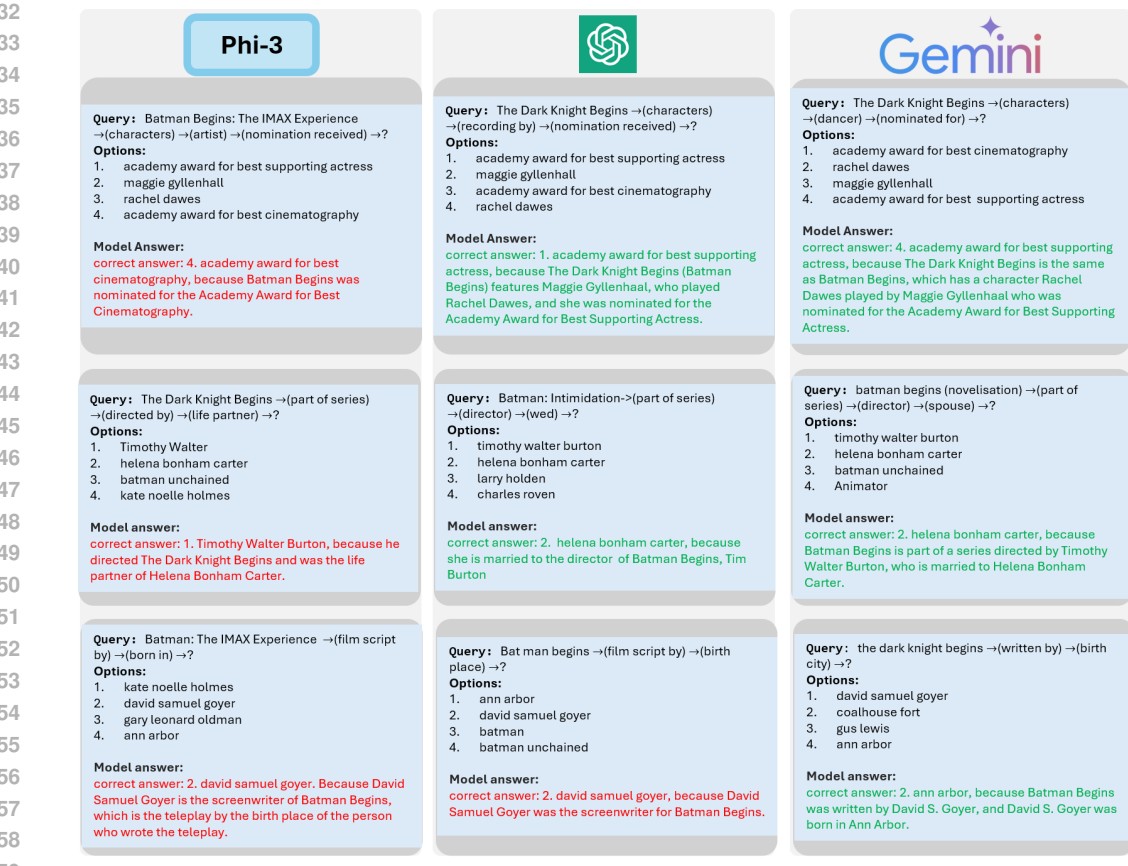

Figure 3: Qualitative analysis of samples used for certifying knowledge comprehension for Vanilla specifications on the Wikidata5m subgraph pivoted at the node for 'Batman Begins' movie. The context provided in the prompts is not shown for brevity. Wrong model responses are colored red and correct ones are colored green. The samples are consistent with our results, wherein Phi-3 (3B) has lower certification bounds than GPT-4o's bounds, which are lower than those for Gemini-Pro.

**Comparison with benchmarking baseline**. Baseline scores of all models consistently approach or surpass the average certification upper bounds, suggesting potential inflation of performance estimates in benchmarking. Contrary to the certification bounds, Mistral-7B significantly outperformed Phi3-3B across all quantizations. Phi3-14B's performance had a substantial decline with 8-bit quantization, far greater than the drop shown by certification. These findings emphasize the need for more reliable and principled evaluation methods grounded with statistical guarantees.

**Quality of bounding intervals**. Table 1 presents average certification bounds over all specifications. A desirable property for the intervals, alongside their high confidence, is that they should be tight, i.e., their range should be small. Tighter intervals indicate precise analysis with less errors. We observe that the average range of the confidence intervals in our experiments is less than $0.12$.

## 4.3 CASE STUDIES

Next, we analyze the certification results, qualitatively. First, we show the responses of 3 models in Figure 3 — Phi-3 (3B), GPT-4o, and Gemini, obtained when certifying them for the Vanilla specification defined over a subgraph pivoted at the node for 'Batman Begins' movie. The samples reflect the certificates. Next, we identify and categorize prominent kinds of model responses. We frequently see the following failure modes — *distracted* and *missed relation*. In the former, the model gets deviated from the query by following the distractor context in its prompt, resulting in an incorrect answer. In the latter, the model skips some reasoning steps needed for the final correct

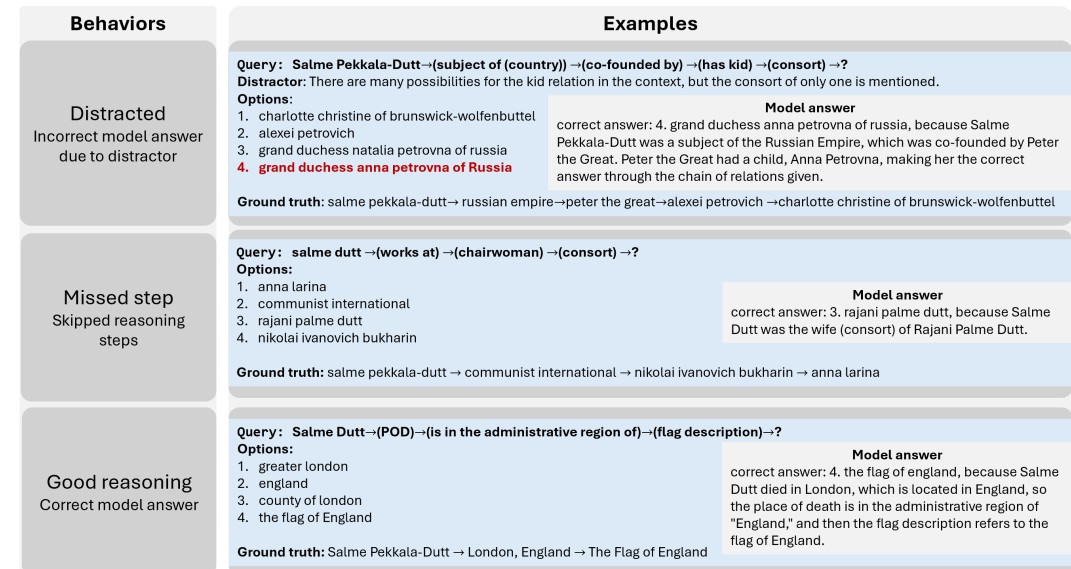

Figure 4: GPT-4o responses showing good reasoning, failures due to distractors or skipped reasoning

answer. In cases of *good reasoning*, model accurately follows the query and gives the correct answer. Figure 4 presents examples of the aforementioned kinds of model responses for GPT-4o.

## 5  RELATED WORKS

**In-context learning**. As LLM context windows increase (Gemini Team, 2024; Chen et al., 2023; Dubey et al., 2024), more information can be provided in the prompts like few-shot demonstrations (Brown et al., 2020) and examples from related tasks (Qu et al., 2024). In-context learning is the emergent behavior (Wei et al., 2022b; Lu et al., 2024) in which LLMs become proficient at a task with demonstrations in prompts. We use in-context learning and few-shot examples to enhance LLMs' knowledge and reasoning capabilities.

**Benchmarking LLM intelligence**. Several benchmarks have been proposed to study the reasoning (Zhou et al., 2022; Huang & Chang, 2023; Plaat et al., 2024; He et al., 2024; Zha et al., 2021), arithmetic (Yuan et al., 2023; Song et al., 2024; Yang et al., 2023a), planning (Pallagani et al., 2023; Valmeekam et al., 2023; Huang et al., 2022), and question-answering (Yang et al., 2018; Ho et al., 2020; Welbl et al., 2018) capabilities of LLMs, which are integral components of human intelligence. These benchmarks provide empirical insights and trends into the performance of LMs. However, these insights are generally for static datasets and are not guaranteed to generalize. On the other hand, certification methods provide guarantees on, for example, the scope (defined by specifications) and confidence of its claims, as we illustrate in this work.

## 6  CONCLUSION AND FUTURE WORK

We present a novel framework to formally certify LLMs for knowledge comprehension. We develop novel specifications that quantify the probability of correct responses over any random knowledge comprehension prompts from distributions derived from knowledge graphs. Certificates consist of high-confidence bounds on the probability of correct knowledge comprehension, thus providing a method to compare different LLMs with formal guarantees. Our experiments show variations in knowledge comprehension along the axes of model size, quantization, and task difficulty. Future work can integrate our framework with knowledge graph construction methods (Ye et al., 2022), to specify and certify LLMs for comprehension and reasoning over less structured/proprietary inputs.

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

# A ABLATIONS

## A.1 FEW SHOT PROMPTS

We conduct an ablation study to examine the impact of varying the number of few-shot examples on Gemini-Flash's performance in the vanilla task setting. While our default configuration uses two few-shot examples, we extend this analysis to include up to five examples. Interestingly, we observe no significant variation in performance across these different few-shot configurations. The results are presented below in 2.

Table 2: Certification results for LLMs in vanilla setting with different number of few-shot examples

| Model | Avg. lower bound | Avg. upper bound | Avg. accuracy |
|---|---|---|---|
| Gemini-1.5-Flash 2Shot (Default) | $0.46 \pm 0.06$ | $0.58 \pm 0.06$ | $0.52 \pm 0.06$ |
| Gemini-1.5-Flash 3Shot | $0.46 \pm 0.06$ | $0.58 \pm 0.06$ | $0.52 \pm 0.06$ |
| Gemini-1.5-Flash 4Shot | $0.46 \pm 0.07$ | $0.58 \pm 0.07$ | $0.52 \pm 0.07$ |
| Gemini-1.5-Flash 5Shot | $0.46 \pm 0.07$ | $0.58 \pm 0.07$ | $0.52 \pm 0.07$ |

## A.2 DISTRACTOR DISTRIBUTIONS

To assess the impact of distractor distribution on model performance, we implement three distinct distractor distribution strategies:

1. Tail-weighted: Linearly increasing weights towards the tail end of the path, prioritizing distractors near the answer node. This serves as our default setting.

2. Head-weighted: Linearly increasing weights towards the head of the path, emphasizing distractors near the query's starting point.

3. Uniform: Equal probability of selecting distractors from any position along the path.

We observe no significant differences in either of the settings. The results are presented in 3 below.

Table 3: Certification results for Gemini-Flash with different distractor distributions

| Model | Avg. lower bound | Avg. upper bound | Avg. accuracy |
|---|---|---|---|
| Gemini-1.5-Flash Setting 1 (Default) | $0.42 \pm 0.10$ | $0.55 \pm 0.10$ | $0.48 \pm 0.10$ |
| Gemini-1.5-Flash Setting 2 | $0.42 \pm 0.11$ | $0.55 \pm 0.11$ | $0.48 \pm 0.11$ |
| Gemini-1.5-Flash Setting 3 | $0.42 \pm 0.11$ | $0.55 \pm 0.11$ | $0.48 \pm 0.11$ |

## A.3 MODEL PERFORMANCES WITH VARYING PATH LENGTH

Among our certificates, we have queries of various lengths. Here we study the effects on models behavior on queries with varying length by considering the number of hops they require to reason to answer the query(which is 1 less than the path length). To do so, we refer to the number of hops to answer a query as k where $1 \leq k < \rho$.

**Varying Setting:** In figure 6 we show plots for various specifications for the GPT4o model.

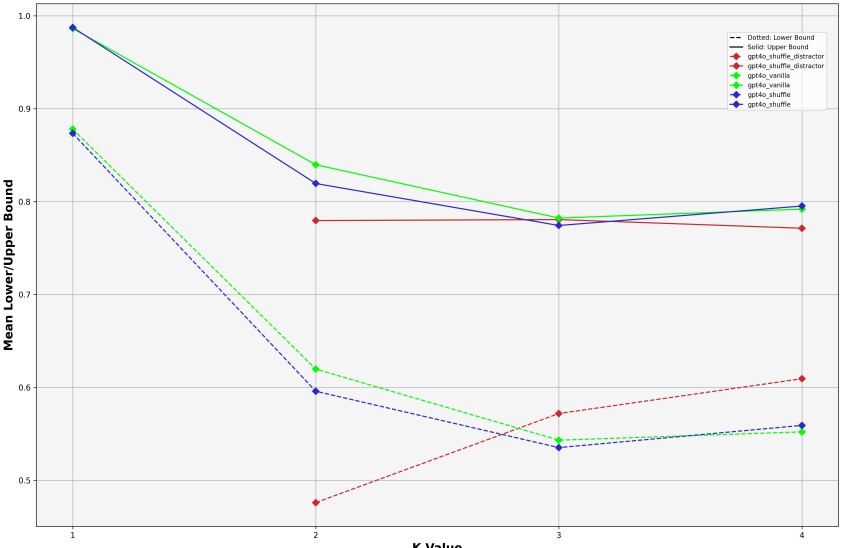

Figure 6: Variations in the bounds against the path lengths across various specifications.

**Varying Quantization:** In figure 7 we show plots when the quantization is varied with the Llama3-8B model on the shuffle specification and their effects on performance.

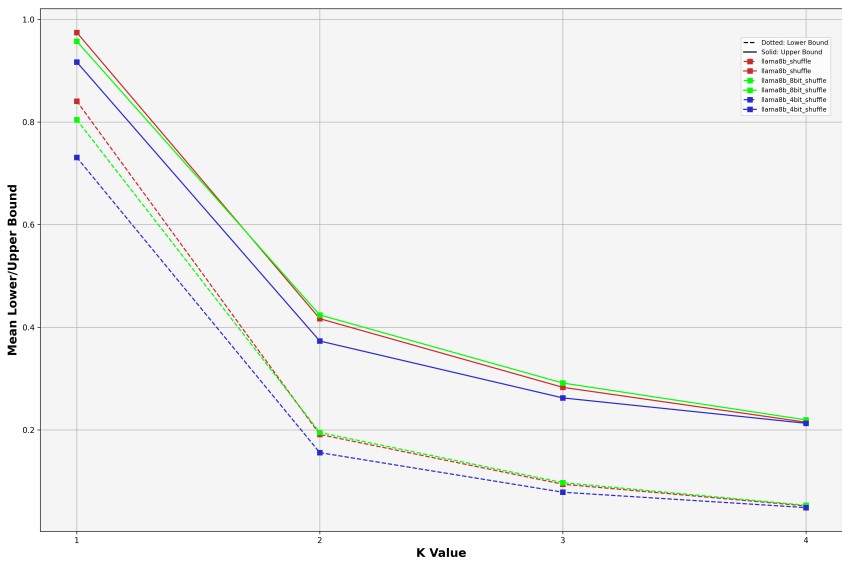

Figure 7: Variations in the bounds against the path lengths across various quantizations.

**Varying Models:** In fig 8 we show plots for the shuffle specification and performance across the models(the open-source models use fp16 precision).

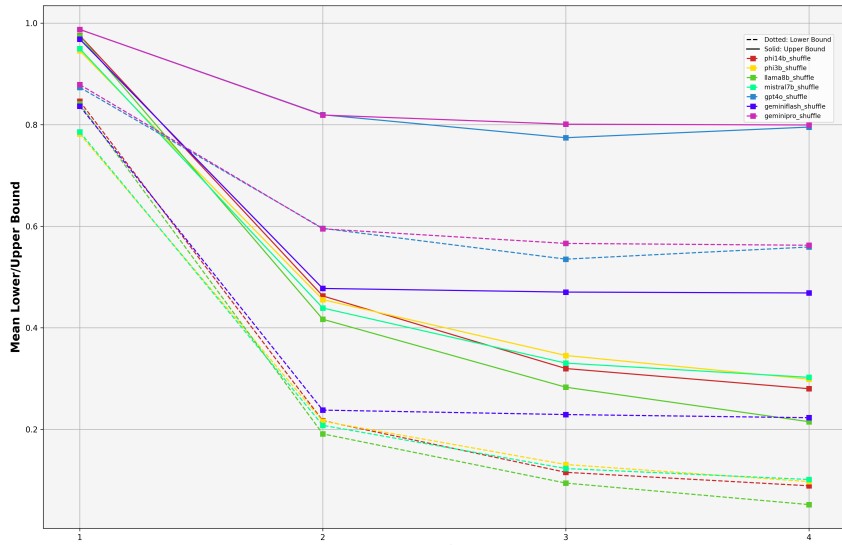

Figure 8: Variations in the bounds against the path lengths across various models in the shuffle setting.

In Figure 6, we observe that the performance across settings converges as k increases and the distractor setting is most impactful on the performance for $k = 2$.

In Figure 7, we infer that as k increases the performance of the models' on the task converges across the different quantizations. We hypothesize this is due to the increasing complexity of the reasoning task.

In Figure 8, we see that larger models (GPT-4o, Gemini-Pro) show less severe drop in performance compared to their smaller models. The figure shows that large models may have learnt to better apply 1-step reasoning for multiple steps when compared to their smaller counterparts.

## A.4 CHAIN OF THOUGHT PROMPTING

We also conduct an ablation on how Chain-of-Thought(COT) prompting can affect the performance of language models on the knowledge comprehension task. Specifically, we investigate the Phi-3 3B model (precision: float16) in the vanilla setting with COT prompting strategy. We augmented our standard few-shot examples ($B.5$) with COT steps and added structured reasoning guidance to the prompt template ($B.3$):

> **COT additions to prompt template**
>
> Answer in the following the below format:
> Let's solve this step by step: 1) Let's identify the starting point and path: - Start: [identify starting entity] - Path to follow: [break down the path components]
> 2) Let's follow the path: Starting from [entity] → [first relationship] → [next entity] → [next relationship] → [next entity] ... [continue as needed]
> 3) Verify our final destination reaches one of the given options
> Therefore, the correct answer is: <option_number>. <option_text>

In the vanilla setting, adding COT prompting improved Phi-3 3B's performance, with the bounds increasing by 0.11 summarized in Table 4. While we acknowledge the potential benefits of COT, earlier experiments were limited due to the significantly increased computational cost (generating 5-8 times more tokens) and the expenses of COT, particularly with closed-source models as output tokens are much more expensive.

Table 4: Certification results for Phi-3 3B with and without COT

| Prompting Strategy | Avg. lower bound | Avg. upper bound | Avg. accuracy |
|---|---|---|---|
| No COT (Default) | $0.34 \pm 0.05$ | $0.46 \pm 0.06$ | $0.40 \pm 0.05$ |
| COT | $0.45 \pm 0.08$ | $0.57 \pm 0.08$ | $0.51 \pm 0.08$ |

## B    KNOWLEDGE GRAPH AND QUERY GENERATION

This section details our experimental setup for generating multi-hop reasoning queries using the Wikidata5m knowledge graph. We describe the structure of the knowledge graph, the process of generating random paths, formulating queries, and creating answer options including distractors.

### B.1    KNOWLEDGE GRAPH STRUCTURE

Our experiments are based on the Wikidata5m knowledge graph (KG). The KG has the following key characteristics:

- **Nodes**: Each node represents an entity and is associated with a text paragraph from Wikidata5m.
- **Edges**: Edges represent relationships between entities.
- **Text Paragraphs**: The text associated with each node may contain information relevant to its connected edges.
- **Node and Edge Aliases**: Each node and each edge has a set of aliases associated with them which are just different names for them.

This structure allows us to generate queries that require reasoning across multiple hops in the graph.

### B.1.1    PREPROCESSING THE WIKIDATA5M KNOWLEDGE GRAPH

To ensure the generation of unambiguous queries and support the certification process, we preprocess the wikidata5m dataset.

1. **Relation Filtering:** We remove relations such as 'instance of', 'subclass of', and 'part of' due to their inherent potential for ambiguity in query formulation.

2. **Relevant Information Extraction for edges:** To ensure the relevance of relationships in the knowledge graph, we require textual evidence for each edge. When entity A is related to entity B, we identify specific sentences in the descriptive text of either entity that explicitly mention any alias of the other entity. We assume these sentences support the relationship's existence. These sentences are then linked to the edge, providing context that can be used to answer queries about the relationship. This approach ensures that the knowledge graph contains valid relationships and the specific text that justifies them, enhancing the available context for further analysis. If we find no supporting text for an edge, we drop that edge from the knowledge graph.

3. **Unicode to ASCII:** For consistency within our experiments, we convert all text containing Unicode characters into their respective ASCII approximations.

### B.2    QUERY GENERATION

We utilize the Wikidata5m knowledge graph for multi-hop query generation. The query generation process involves the following steps:

### B.2.1    RANDOM PATH GENERATION

We begin by selecting a pivot node $v_0$ in the knowledge graph $\mathcal{G}$. From this pivot, we construct a local subgraph $\mathcal{G}(v_0)$ consisting of all paths $\Pi_{v_0}$ originating from $v$. This local subgraph serves

as the domain for our path generation process. As arbitrary long paths can lose their semantic meaningfulness, we use a constraint $\rho$ to restrict the length of paths from the pivot node in the subgraph to be maximum $\rho$.

Within $\mathcal{G}(v_0)$, we generate a path $\Pi$ using a randomized depth-first search algorithm. The length of this path, denoted as $k_{choice}$, is sampled randomly from the set $1, 2, ..., \rho$ according to a discrete uniform distribution.

This randomized depth-first search traverses the neighbors of each node in $\mathcal{G}(v_0)$ in a random order, which directly corresponds to the sampling process described in line 10 of Algorithm 1. Specifically, at each step, we sample the next node in the path from a discrete uniform distribution over the current node's neighbors within the local subgraph, expressed as $\sim (\mathcal{D}([v' \mid (v_i, v') \in \mathcal{E} \land v' \in \mathcal{G}(v_0)]))$, where $v_i$ is the current node in the path.

To ensure well-defined queries with unique answers, we introduce an additional constraint on path generation. This constraint requires that each generated path be unique in terms of its sequence of relationships. Specifically, traversing the path from the initial node using the specified relations must lead to a single, unambiguous answer node. This approach prevents queries with multiple valid answers, which would complicate the evaluation of the language model's performance. It's important to note that this uniqueness constraint applies only to the specific path being generated. Nodes within the path may still have multiple edges with the same relation type to other entities not on the path. This allowance maintains the natural complexity of the knowledge graph structure, where entities can have multiple relationships of the same type with different entities.

The pseudocode for the path generation algorithm is specified in 2

---

**Algorithm 2** Random Path Generation

---

1: **Input:** Graph $G$, Integer $k$, Vertex $source$
2: **Output:** $path$
3: $path\_len \leftarrow$ **RandomInteger**$(1, k)$
4: $path \leftarrow$ None
5: **while** $path$ is None **do**
6:    $path \leftarrow$ **DFSPath**$(G, source, path\_len)$
7:    **if** not **IsUnique**$(path)$ **then**
8:       $path \leftarrow$ None
9:    **end if**
10: **end while**
11: **return** $path$

---

### B.2.2 QUERY FORMULATION

Once a valid path $\Pi$ is generated, we convert it into a query string. This process aligns with line 11 in Algorithm 1. The query is constructed by sampling aliases for each node and relation in the path. For example, a path $\Pi = [A, B, C]$ might be converted to a query "sampled_alias(A) $\rightarrow$ sampled_alias((A, B)) $\rightarrow$ sampled_alias(B) $\rightarrow$ sampled_alias((B, C)) $\rightarrow$ ?". Here the tuple of two nodes represents their edge. The aliases are sampled randomly from a discrete uniform distribution over the available aliases for a node or an edge.

### B.2.3 EXAMPLE QUERY GENERATION

To illustrate our query generation process, consider the scenario of a path in our subgraph as shown in 9.

Our algorithm would construct the following query from the path presented in 9:

"Chandler Bing$\rightarrow$(actor)$\rightarrow$(birth_date)$\rightarrow$?"

This query requires the LLM to reason through two hops in the knowledge graph:

1. Identify the actor who played Chandler Bing (Matthew Perry)

2. Find the birth date of Matthew Perry (19 August 1969)

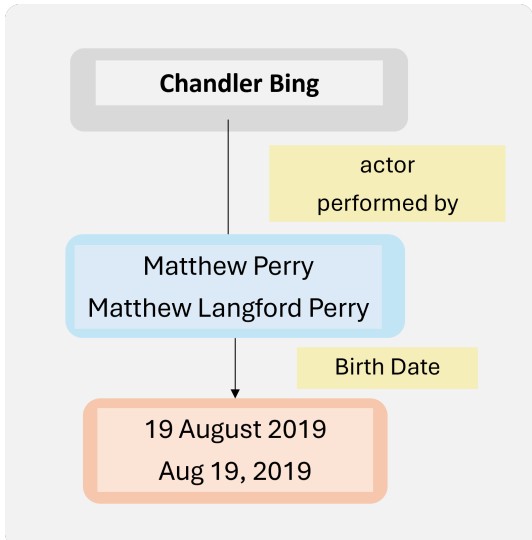

Figure 9: Potential Path in a Subgraph where Pivot is Chandler Bing

This example demonstrates how our query generation process creates questions that require multi-hop reasoning, leveraging the structure and relationships within the knowledge graph.

### B.3 PROMPT CONSTRUCTION

The final prompt is constructed using a template applied to the query. This process involves several steps, each addressing specific requirements:

- Query Formulation: Convert the generated path into a query string as described earlier.
- Context: This is the supporting text we provide the LLM to answer the query correctly. We additionally trim the context to fit within the LLM's context length limits.
- Few-shot Examples: Include examples to guide the LLM in understanding the query format and expected answer structure.
- Answer Options Generation: Create a set of possible answers, including the correct one. The LLM has to choose one of these options as the correct one.
- Distractors: In the distractor setting, we need to find distractors for the query which need to be included in the prompt.

These inclusions ensure that the prompt is comprehensive, fits within model constraints, and provides sufficient guidance for the LLM to generate accurate responses. We also provide information on the aliases used and the entities they correspond to in the prompt, to ensure that the LLM knows about the alias.

### B.3.1 DISTRACTOR SELECTION

Distractors are crucial in assessing an LLM's comprehension and reasoning abilities. We hypothesize that distractors to nodes later in the path closer to the answer would provide more difficulty for the LLM due to their proximity to the answer node. Our distractor selection process, implemented in Algorithm 4, employs a weighted sampling approach to prioritize distractors associated with entities closer to the path's end. The algorithm first identifies all potential distractors for each node in the path, then assigns weights inversely proportional to their distance from the final answer node. This weighting scheme favors distractors linked to nodes near the path's end, but still allows for the selection of distractors related to any node in the path. This refers to the sampling procedure in the line 12 of the probabilistic progam specification 1. By performing weighted sampling from this pool, we ensure a balance between highly relevant distractors and a diverse selection across the entire path.

B.3.2 ANSWER OPTIONS

After formulating the query, we generate a set of answer options. This set includes:

- The correct answer: The last entity in the generated path.

- Other entities in the path.

- Related entities: Entities that share some edge with nodes in the path but are not part of the path.

- Distractors: A distractor is a node in the knowledge graph $\mathcal{G}$ that shares a relation with a node in the path, mirroring the relation that continues the path, but the distractor is not itself part of the path. For a formal definition, refer to Definition 3.3. These are only included in the options in the distractor setting.

The process of generating answer options is detailed in Algorithm 3. In the algorithm, we sample answer options from the set described above so we are basically sampling from the nodes as in the probabilistic program specifiction line 12 1. The answer option algorithm assumes that distractors are input in a list according to the order of preference.

---

**Algorithm 3** Generate Answer Options

1: **Input:** $correct\_ans$, $distractors$, $path\_entities$, $random\_entities$, $Graph$, $min\_num\_options$
2: **Output:** $options$
3: $options \leftarrow [(correct\_ans] \cup distractors$
4: Add path entities to $options$
5: Add random entities from $random\_entities$ to $options$
6: **return** **Shuffle**($options[: min\_num\_options]$)

---

**Algorithm 4** Get Best Distractor

1: **Input:** Graph $G$, Path $\Pi$
2: **Output:** $best\_distractor$
3: $D \leftarrow []$ {List of distractors}
4: $W \leftarrow []$ {Weights for distractors}
5: **for** $i \leftarrow 0$ to $\text{len}(\Pi) - 2$ **do**
6:    $v \leftarrow \Pi[i]$
7:    $N \leftarrow \text{GetNeighbors}(G, v)$
8:    $N\_distractors \leftarrow \text{FilterDistractors}(N, v, \Pi)$
9:    Extend $D$ with $N\_distractors$
10:   Extend $W$ with $[i + 1] * len(N\_distractors)$
11: **end for**
12: **if** $D$ is not empty **then**
13:    **return** **WeightedRandomChoice**($D, W$)
14: **else**
15:    **return** None
16: **end if**

---

B.4 CONTEXT TRIMMING

To address the input length limitations of various LLMs, we implement a context trimming procedure. Including all text associated with each node in a reasoning path can result in excessively long contexts. Our procedure aims to preserve the most relevant information from the knowledge graph and supporting texts while respecting each model's maximum input length. This involves identifying relevant sentences per edge in the graph and then trimming the context for each query based on this information.

### B.4.1 FINDING RELEVANT SENTENCES PER EDGE

Each node in the Wikidata5m knowledge graph has associated textual support for its relations. We utilize this textual information to provide query-relevant context. We need to determine the relevant information from the textual supports for each edge as this would help us trim the contexts accordingly. For each edge $(u, v)$ in the knowledge graph used in the query or answer options generation, we perform the following steps:

1. **Collect Aliases and Text:** We gather aliases and the associated text paragraphs for both nodes $u$ and $v$.

2. **Split into Sentences:** We split the text paragraphs of $u$ and $v$ into individual sentences using NLTK.

3. **Identify Relevant Sentences:** We identify sentences that explicitly link the two nodes. A sentence from $u$'s text is considered relevant if it contains an alias of $v$, and vice versa.

4. **Discard Edges without Relevant Sentences:** If no relevant sentences are found for an edge, it is deemed unsupported and is discarded from the graph.

5. **Prepend First Sentence:** To ensure the entity's primary name or common alias is included, we prepend the first sentence of each node's text to its list of relevant sentences.

### B.4.2 TRIMMING TO FIT CONTEXT LENGTH

When constructing the final prompt for the LLM, we prioritize including the most relevant information within the model's context length limit. Therefore we need to trim the context according to the LLM's context limit. We use the following procedure (detailed in Algorithm B.4.2):

1. **Create Sentence Lists:** We create three lists of sentences:
   - $S_{all}$: Contains all sentences from the text paragraphs of nodes involved in the query and answer options.
   - $S_{query}$: Contains all relevant sentences for the edges that constitute the query path.
   - $S_{options}$: Contains all relevant sentences for the edges used to generate the answer options.

2. **Construct the Final Context:**
   (a) We prioritize including all sentences from $S_{query}$ as they are directly related to the query.
   (b) Next, we add as many sentences from $S_{options}$ as possible, given the remaining context length limit.
   (c) Finally, we fill the remaining space with sentences from $S_{all}$ that have not been included yet, ensuring no sentence is repeated.

---

**Algorithm 5** Context Construction

---

1: **Input:** $S_{all}, S_{query}, S_{options}, L_{max}$
2: **Output:** $C_{trimmed}$
3: $C \leftarrow S_{query}$
4: **ASSERT** TokenizedLength$(C) \leq L_{max}$
5: $S_{seen} \leftarrow$ UniqueSet$(C)$
6: **for** each $s$ in $S_{option} + S_{all}$ **do**
7:    **if** $s \notin S_{seen}$ **and** TokenizedLength$(C + s) \leq L_{max}$ **then**
8:       $C \leftarrow C + s$
9:       Add $s$ to $S_{seen}$
10:    **end if**
11: **end for**
12: **return** $C$ as $C_{trimmed}$

---

## B.5 FEW-SHOT EXAMPLES

To guide the LLM towards the desired response format and demonstrate the reasoning process, we include 2 few-shot examples in the prompt. These examples provide a clear illustration of how to approach the multi-hop reasoning task.

We use the following few-shot examples:

---

**Few Shot Examples**

**Common Context:** entity_B is the son of entity_A. entity_E is the sister of entity_A. entity_B leads entity_C. Entity_D is a member of Entity_C. Entity_D is a friend of entity_E. entity_E has mother entity_F who likes the services of entity_C.

**Question 1:** entity_A → (father of) → (leader of) → ?
**Options:** 1. entity_F, 2. entity_C, 3. entity_D, 4. entity_E, 5. entity_B
**Answer:** 2. **entity_C**
**Explanation:** entity_A → (father of) entity_B → (leader of) entity_C
*How to get answer:* Find who entity_A is father of to get entity_B, then find what B is the leader of to get entity_C.

**Question 2:** entity_B → (chief of) → (constitutes) → (companion of) → ?
**Options:** 1. entity_F, 2. entity_C, 3. entity_D, 4. entity_E, 5. entity_A
**Answer:** 4. **entity_E**
**Explanation:** entity_B → (chief of) entity_C → (constitutes) entity_D → (companion of) entity_E
*How to get answer:* Find what entity_B is the chief of to get entity_C, find what entity_C constitutes to get entity_D, then find the companion of entity_D to get entity_E.

---

## B.6 FINAL PROMPT

The final prompt presented to the LLM is constructed using a template that incorporates several key elements:

**Trimmed Context [B.4]:** The relevant context extracted and trimmed.

**Query [B.2]:** The multi-hop query.

**Answer Options [B.3.2]:** The generated answer options, including the correct answer and distractors.

**Few-Shot Examples [B.5]:** A set of examples demonstrating the desired response format.

The prompt template is structured as follows:

---

**Prompt Template**

{few_shot_examples}
Actual Query: Given Context: {context}
Answer the question: {query}
answer the question by selecting the correct answer from the following options:
{options}

The format for beginning your response is:
correct answer: $< option\_number >$ . $< answer >$, $because < succinct\ reason >$
follow this exact format and only choose from the given options

---

**Estimating the number of unique prompts:** We estimate a lower bound on the number of unique prompts that can be generated from the Wikidata5m Knowledge Graph (KG) by quantifying the

potential unique queries within the graph. Each query can be formulated into multiple prompts through variations in answer presentation, thus making query count a conservative estimate. We analyzed the 50 subgraphs employed in our experiments. For each subgraph, we calculated the number of unique paths(upto the maximum path length hyperparameter, $\rho = 4$) and calculated the number of possible queries for each path using the number of aliases for each each entity and relation within a path. This analysis provides an estimate of the unique query generation capacity inherent in subgraphs in our KG.

The mean number of unique queries was $3.04 \times 10^{15}$ with a median of $1.24 \times 10^{15}$. The minimum and maximum observed values were $1.36 \times 10^{12}$ and $1.46 \times 10^{16}$, respectively.

Importantly, these figures conservatively estimate the number of unique prompts, as they only consider query variations and not the diversity introduced by different answer options. The actual number of unique prompts is likely significantly larger, making exhaustive enumeration of all possible generated prompts infeasible.

## B.7 RESPONSE CHECKER FUNCTION

We implement a simple response checker function to evaluate the correctness of the model's answers. The function is defined in Algorithm B.7. We write a regular expression to account for trivial formatting errors like extra spaces, brackets, incorrect punctuation, etc.

---

**Algorithm 6** Response Checker

---

1: **Input:** $model\_answer$, $correct\_answer\_num$
2: **Output:** $is\_correct$
3: $model\_answer \leftarrow$ LowerCase($model\_answer$)
4: $correct\_answer\_num \leftarrow$ LowerCase(ToString($correct\_answer\_num$))
5: $pattern \leftarrow$ SpecializedRegularExpression("correct answer: " + $correct\_answer\_num$)
6: **if** RegexMatch($pattern$, $model\_answer$) **then**
7:    $is\_correct \leftarrow 1$
8: **else**
9:    $is\_correct \leftarrow 0$
10: **end if**
11: **return** $is\_correct$

---

