# OpenReview forum: "Decoding Intelligence: A Framework for Certifying Knowledge Comprehension in LLMs"
_ICLR.cc/2025/Conference — Submitted to ICLR 2025_

### Official Review · Reviewer_mSM5 · 2024-10-30

**Soundness:** 3
**Presentation:** 3
**Contribution:** 2
**Rating:** 6
**Confidence:** 3

**Summary:**

The paper introduces QuaCer-C, a protocol to assess the knowledge comprehension abilities of LLMs, i.e. their ability to extract information from reference inputs and reason over it to answer questions. To this end, the authors introduce an evaluation protocol that constructs multi-step reasoning queries together with multiple-choice answers and gathers reference information based on traversing a knowledge graph. By sampling many paths starting from the same root node, the authors are able to estimate confidence intervals for whether an LLM will answer knowledge comprehension queries based on that root node correctly. The paper uses this approach to quantitatively and qualitatively evaluate the knowledge comprehension abilities of several popular open-weight as well as closed models. The results show that closed models significantly outperform closed ones, and sheds lights on the failure modes of knowledge comprehension.

**Strengths:**

1. Overall, the paper is well-written and easy to follow.
2. I believe that the introduced knowledge comprehension test can serve as a useful benchmark for evaluating the ability of LLMs to retrieve information from prompts, reason about that information and use it to answer complex questions.
3. The paper provides a comprehensive assessment of the knowledge comprehension abilities of many of the currently most popular LLMs.

**Weaknesses:**

1. I am not sure in which situations the correctness certificates derived by QuaCer-C would be useful. The certificates hold for prompts sampled from the same distribution as the 250 sample prompts. But that means that certificates can only be obtained for cases where a corresponding knowledge graph exist and the relevant queries can be expressed as graph traversals. But in such cases, it would be much simpler to query the knowledge graph directly, rather than using an LLM to extract information from and reason over it. The cases where LLM-based knowledge comprehension is actually required are typically much less structured documents without a corresponding knowledge graph, but in those cases QuaCer-C cannot compute certificates. I still think the prompt construction and evaluation approach can serve as a useful benchmark for the knowledge comprehension abilities of LLMs, but I don't see a scenario where the derived certificates would be useful.
2. The approach might incorrectly mark answers as wrong in case of 1 - n or m - n relationships. E.g. in Figure 4, first row, in the example "Batman Begins: ... -> (characters) -> (artist) -> (nomination received) -> ?" there could be multiple characters (1 - n relationship) whose artists might have received different nominations. The model might pick a different but valid character than intended and then reason correctly, potentially using its parametric knowledge, and arrive at a different than expected, but still correct answer.
3. The paper claims that larger models are better at knowledge comprehension. While Table 1 provides some evidence in this direction, I believe that it is insufficient to confidently claim a size-dependent relationship, because of a number of potential confounders: 1) The smaller models in the table are all open source ones, while the larger ones are closed, and their size is not (officially) known. 2) Except for (Phi-3-3B, Phi-3-14B) and (Gemini-1.5-Flash, Gemini-1.5-Pro) (whose size difference and other potential differences are unknown), all models belong to different families, so other than size they might also differ in training data mixtures, training strategies and architecture. The only really comparable datapoint here is (Phi-3-3B, Phi-3-14B), and those two models show 1% or less difference. To make claims about model size effects more reliable, comparisons of several (open) models within the same family would be needed.
4. Minor clarity issue: It was not clear to me how the few-shot examples are constructed until I came across Appendix B.5. Please reference the appendix in the main paper and provide some minimal information about the few-shot examples, e.g. that the same fixed set of examples is used for all prompts.

**Questions:**

### Questions
1. Does the approach also work without few-shot examples? Or are they needed to convey the answer format?
2. How long is the context (per node) and does it contain information beyond the Wikipedia page's abstract? The authors mention that only one distractor is included due to context size limitations. However, all the studied models support, or have versions that support, context lengths of at least 32k. At least if Llama-3.1-8B was used, rather than Llama-3-8B, which I'm not sure about.
3. Do the authors think that techniques like chain-of-thought prompting would change the results? The paper investigates problems that inherently require multi-step reasoning, whereas the evaluation expects models to produce the answer almost as the first token. Allowing for additional reasoning steps might significantly improve accuracy.

### Suggestions
1. It would be helpful to include the appendix into the main paper's PDF, not as a separate file.
2. It might be helpful to name the certified property, i.e. "our overall property", line 228.
3. Currently, prompts are constructed by sampling graph trajectories starting from a specific root node. Another interesting approach might be to sample trajectories whose edges all have the same relationship type, e.g. which are all of the form "... -> (appeared in movie) -> (directed by) -> ?", irrespective of the nodes that appear in them. Such an approach might be able to assess/certify how well a model can comprehend knowledge about a particular multi-step relationship, irrespective of what the exact entities are, e.g. how well the model can comprehend which directors an actor worked with.

---

> ### Author Response · Authors · 2024-11-21
>
> We thank the reviewer for their time and constructive feedback. We address their concerns below. We hope our response mitigates their concerns and they consider increasing their support for our paper.
> > Utility of certificates
>
> The reviewer correctly identifies that certification guarantees generalize only over the distribution given in the specification. Certification is generally useful as:
> - Certificates from traditional neural network certification methods [1,2] are used to quantify model robustness as number of specifications that could be certified (deterministic certification) for them. We envisage similar utility for QuaCer-C as well. Average certification bounds can be used to assess and compare the general knowledge comprehension capabilities across LLMs.
> - Even with knowledge graphs, language models (LMs) are needed to effectively parse natural language prompts, extract entities/relations involved in queries, and respond in desired output format. LMs enable doing this seemlessly, without significant manual efforts. Hence, they may be preferable in natural language question answering settings, even when knowledge graphs are available.
> - As mentioned in our future work, for domains with documented knowledge but no knowledge graph, existing knowledge graph construction methods [3] can be integrated with QuaCer-C to certify LLMs for knowledge comprehension. Our work, being the first step in this direction, provides one component of such a pipeline.
> > Instances with multiple correct answers
>
> We acknowledge this possibility and hence we prompt with multiple-choice questions (MCQs), where only 1 answer is correct and we evaluate model's answer with the known correct answer. Generalizing beyond MCQs requires evaluators that can check for any possible correct answer (and their aliases) in LLM's response. Developing such evaluators is complementary to our research and our framework can easily incorporate them. As we are not aware of any reliable evaluators with low false evaluation rates, we conduct experiments with MCQs. Our theoretical framework, however, generalizes beyond MCQs, to free-form QA with multiple possible answers (similar to aliases).
> > Certifying more models with varying sizes in same family
>
> We thank the reviewer for the suggested experiment. We show results in the "new experiments" section of the general response. We understand the importance of other factors such as training data in LLM performance, and acknowledge them alongside model size in Section 4.3.
> > Constructing few-shot examples
>
> We have updated the main paper with a reference to Appendix B.5 on few-shot examples. We use the same few shot examples across all prompts and have updated this detail in line 323.
> > Need for few-shot examples
>
> QuaCer-C theoretically does not require few shot examples in prompts. It can work, given an evaluator for LLM responses that can extract the correct answer from unstructured text. However, such evaluators tend to be quite inaccurate in our experience and also observed by prior works [4]. Hence in practice, as correctly identified by the reviewer, we need few shot examples to convey the structure of the query and expected response to LLMs. This enables using string matching to automatically and accurately extract the LLM's response from the generated text and evaluate it.
> > Length, contents of context per node?
>
> Context per node contains ~300 tokens consisting of only the abstracts of Wikipedia pages of the node's entity.
> > Limitation in number of distractors due to context length.
>
> We certify several models, some of which have small context windows. These include Llama-3-8B (not 3.1) and Mistral-7B with 8k tokens each. As we want to compare the performance of different models on the same standards, we restrict to only 1 distractor in our experiments. We also want to maintain a low proportion of distracting information, relative to useful information. As the certification also uses samples with shorter path length, e.g., 3 nodes, restricting to 1 distractor ensures that the prompt has reasonable complexity. Note, however, our framework is flexible to allow multiple distractors as well, as required by the use case.
> > Use of Chain-of-Thought (COT)
>
> We agree with the reviewer that COT may affect LLM performance. Please check our "new experiments" section in the general response for our certification results with COT (also in Appendix A.4).
> > Certification with edges with same relations.
>
> We thank the reviewer for their suggestion. We show certificates for the suggested specifications in "new experiments" in the general response.
> ## References
> 1. AI2: Safety and Robustness Certification of Neural Networks with Abstract Interpretation, Gehr et al., 2018
> 2. Certified Adversarial Robustness via Randomized Smoothing, Cohen et al., 2019
> 3. A Comprehensive Survey on Automatic Knowledge Graph Construction, Zhong et al., 2023
> 4. Judging LLM-as-a-Judge with MT-Bench and Chatbot Arena, Zheng et al, 2023

---

> ### Comment · Reviewer_mSM5 · 2024-11-27
>
> I would like to thank the authors for their detailed response, for providing clarifying information, and for conducting additional experiments addressing my concerns and questions.
>
> ### Utility of certificates
>
> Being able to specify knowledge comprehension queries over knowledge graphs in natural language (NLQs) and using LLMs to interpret them seems useful.
> However, if accurate responses are important, the more reasonable approach seems to be to use LLMs to translate the NLQs into a formal query language, execute them using some form of graph algorithm, and then process the results using LLMs again.
> This would be analogous to using LLMs as a translation layer to SQL for databases, rather than letting LLMs interpret table data directly.
> If accuracy is not critical, then certificates would probably not be required.
> Therefore, it is still not clear to me in which scenarios the derived certificates would be useful.
>
> ### Additional results
>
> The additional results are interesting and reassuring.
> The statements about the effects of model size are more convincing to me now.
> Seeing such a large effect from CoT prompting is also striking.
>
> I find the results on paths with fixed sequences of relations to be promising as well, since I find measuring and potentially certifying knowledge comprehension abilities over these composite relations potentially more useful than for relation chains starting from the same node.
> I believe that a future version of the paper could benefit from a stronger focus on these types of relations.
>
> I also think the results on data with random labels (suggested by reviewer 9uCh) are valuable, since they disentangle the models' knowledge comprehension abilities from confounding parametric knowledge.
>
> ---
>
> While I believe that the new results are valuable, I think that the paper needs another iteration to incorporate them properly.
> My main concern about the utility of the certificates also remains.
>
> Therefore, I will maintain my score.

---

> ### Author Response · Authors · 2024-12-01
>
> We thank the reviewer for their feedback and appreciation of our additional results. We are constantly endeavoring to improve our work and are grateful for the enhancements suggested by the reviewer. We want to clarify our position on the utility of the certificates in this response.
>
> We think that the alternative way of using LLMs with knowledge graphs suggested by the reviewer is interesting. However, our setting of using LLMs for end-to-end question-answering is conventionally popular among works on reading comprehension [1-3]. Moreover, our framework can be trivially extended to certify LLM systems (like the one proposed by the reviewer comprising of the LLM as a parser and a knowledge graph querying engine), as we just assume query access to the question answering system. Traditionally, benchmarking datasets to study knowledge comprehension [3-7] have also been developed with knowledge graphs, similar to our use of knowledge graphs for specifying correct knowledge comprehension. Hence, we believe that this setting is important and certificates for it are useful evaluations of knowledge comprehension by LLMs or LLM systems.
>
> > If accuracy is not critical, then certificates would probably not be required.
>
> We respectfully contradict the reviewer. Accuracy is important for knowledge comprehension, as otherwise the task is pointless. LLMs have been conventionally compared based on their knowledge comprehension accuracy and leaderboards have been designed for the same [8,9]. Hence, accuracy is critical and certification is a reliable way to assess knowledge comprehension performance of LLM-based question answering systems.
>
> ## References
> 1. HOTPOTQA: A Dataset for Diverse, Explainable Multi-hop Question Answering, Yang et al., 2018.
> 2. A Survey on Machine Reading Comprehension: Tasks, Evaluation Metrics and Benchmark Datasets, Zeng et al., 2020.
> 2. Constructing Datasets for Multi-hop Reading Comprehension Across Documents, Welbl et al., 2018.
> 3. KEPLER: A Unified Model for Knowledge Embedding and Pre-trained Language Representation, Wang et al., 2021.
> 4. Multimodal Analogical Reasoning over Knowledge Graphs, Zhang et al., 2022.
> 5. Variational Reasoning for Question Answering with Knowledge Graph, Zhang et al., 2017.
> 6. OpenDialKG: Explainable Conversational Reasoning with Attention-based Walks over Knowledge Graphs, Moon et al., 2019.
> 7. https://crfm.stanford.edu/helm/classic/latest/#/groups/natural_qa_openbook_longans
> 8. https://paperswithcode.com/task/reading-comprehension

---

> > ### Author Response · Authors · 2024-12-02
> >
> > We would like to add on to our justification on the utility of the certificates by highlighting recent work from Anthropic. [1] recommends using statistical methods like ours, over standard evaluations. Given that prior evaluations have been done over datasets from knowledge graphs, certifying specifications defined with knowledge graphs can be useful assesssments of the knowledge comprehension capabilities of LLMs.
> >
> > ## References
> > 1. Adding Error Bars to Evals: A Statistical Approach to Language Model Evaluations, Miller et al., 2024.

---

### Official Review · Reviewer_w1nm · 2024-11-03

**Soundness:** 2
**Presentation:** 2
**Contribution:** 2
**Rating:** 3
**Confidence:** 4

**Summary:**

This paper aims to develop a formal certification framework for evaluating knowledge comprehension in LLMs. The authors propose an approach that frames knowledge comprehension as a formal specification using a knowledge graph. The accuracy of LLM responses is assessed by the probability of generating correct answers to knowledge comprehension prompts sampled from a distribution based on the knowledge graph. However, this approach, in my opinion, closely resembles a basic KBQA evaluation process for LLMs and lacks difference compared to existing work. Furthermore, current proprietary models, such as Gemini-Pro and GPT-4o, have already demonstrated impressive accuracy in knowledge utilization, as shown in Figure 3, with performance scores between 0.7 and 0.8, which raises questions about the significance of this task and the motivation of this work.

**Strengths:**

1. This paper provides a detailed description of the approach, including the formalization, theoretical framework, and algorithmic implementation.
2. Models of varying sizes and employing different pretraining techniques are evaluated.

**Weaknesses:**

This paper provides an extensive and complex introduction and description of the approach for formalizing knowledge comprehension evaluation using a knowledge graph. The knowledge comprehension capability of LLMs is assessed by measuring the accuracy of their responses to prompts sampled from paths within the knowledge graph. However, (1) there is no rigorous theoretical proof to guarantee the approach, and (2) it appears to be a very basic, standard KBQA evaluation process using LLMs nowadays, lacking distinction from existing work. I find the motivation, novelty, and differentiation of this work unclear. Some related work is omitted like [1,2]

[1] Zha, Hanwen, Zhiyu Chen, and Xifeng Yan. "Inductive relation prediction by BERT." Proceedings of the AAAI conference on artificial intelligence. Vol. 36. No. 5. 2022.
[2] He, Xiaoxin, et al. "G-retriever: Retrieval-augmented generation for textual graph understanding and question answering." arXiv preprint arXiv:2402.07630 (2024).

**Questions:**

1. Is the evaluation process merely a standard method of sampling questions from a knowledge graph to assess LLMs? If so, why not utilize existing KBQA/GraphQA datasets?
2. Is there any theoretical guarantee for the bounds introduced?

---

> ### Author Response · Authors · 2024-11-21
>
> We thank the reviewer for their time and constructive feedback. We address their concerns below. We hope our response mitigates their concerns and they consider increasing their support for our paper.
> > Differences from existing works on KBQA and why we can't use standard benchmarks.
>
> Please refer to general response.
>
> > Significance of task and motivation of work.
>
> Knowledge comprehension is an important evaluation task for human learners. As LLMs attempt to achieve human-level intelligence, they should be capable to excel at knowledge comprehension, i.e., attain high performance scores (approaching 100%). Even if we consider the proprietary models such as Gemini-Pro and GPT-4o which achieve high performance, they exhibit several instances of failed knowledge extraction and/or reasoning. For example, Figure 4 shows an example of failed reasoning in GPT-4o. Table 1 shows the reduction in the knowledge comprehension performance of Gemini-Pro and GPT-4o, when the information is shuffled and distractors are included in the prompt (Shuffle Distractor setting), thus indicating that these models are not effectively able to remove additional, distracting information and navigate through shuffled pieces of information, which can be generally done by humans. Hence, knowledge comprehension is not a solved problem and we need reliable assessment and enhancement of this capability in LLMs. Prior works [1,2] have also extensively benchmarked LLMs on knowledge comprehension. However, our work differs from them by providing a reliable certification method for this property.
>
> > Adding suggested references
>
> We thank the reviewer for the references. We have included them in our updated related works section.
>
> > Theoretical guarantees on bounds.
>
> The certification bounds are such that the true probability of correct response for any prompt in a given distribution (e.g., distribution in lines 1-5 of Algorithm 1) $p^*$ is within the bounds with high-confidence. That is, for bounds $[l,u]$, $Pr[p^*\in[l,u]]\geq 1-\delta$, where $\delta>0$ is a small, user-specified constant. This is a property of the Clopper-Pearson confidence intervals [3], which we use as certification bounds. We provide these details in Section 3.2 of the paper. Benchmarking over static datasets, on the other hand, does not give any guarantees on the generalization of the results.
>
> ## References
> 1. Large Language Models' Expert-level Global History Knowledge Benchmark (HiST-LLM), Hauser et al., 2024.
> 2. DOCBENCH: A Benchmark for Evaluating LLM-based Document Reading Systems, Zou et al., 2024.
> 3. The Use Of Confidence Or Fiducial Limits Illustrated In The Case Of The Binomial, Clopper and Pearson, 1934.

---

### Official Review · Reviewer_9uCh · 2024-11-04

**Soundness:** 3
**Presentation:** 3
**Contribution:** 2
**Rating:** 6
**Confidence:** 3

**Summary:**

The authors introduce QuaCer-C, a framework designed to assess knowledge comprehension in LLMs by sampling paths of lengths 1 to 5 from the WikiData5m knowledge graph and constructing context + distraction + query sets as tasks for the models to complete. Since responses are evaluated on a success/fail binary basis, Clopper-Pearson confidence intervals are used to establish upper and lower bounds for the resulting metrics. Experiments on major closed-source and open-source LLMs indicate that larger models perform better, while shuffled contexts and added distractors degrade performance.

**Strengths:**

1. Provides a robust quantitative probabilistic framework for evaluation.
2. Overall, the presentation is clear and the structure flows well.
3. Accompanied by code for reproducibility.

**Weaknesses:**

1. The findings are somewhat predictable and could benefit from deeper insights.
2. Some redundant content in the main text could be replaced by key details currently in the Appendix, such as the process for generating queries and context.
3. There’s ambiguity as to whether the LLM’s responses are derived from embedded knowledge or the provided context, thus raising questions about true comprehension. The prompt itself does not restrict the LLM to answer based solely on the given context. For example, in a hypothetical question like “Matthew Perry→(character_acted_by)→(birth date)→?”, the LLM could respond from its internal knowledge base rather than relying solely on the provided context.

**Questions:**

1. Given that the framework already uses a probabilistic approach, why not leverage the fact that an LLM can act as an implicit conditional probabilistic model? For instance, adjusting the output threshold or re-querying could yield probabilities that reflect comprehension more accurately.
2. Relating to weakness 3: Why are aliases for entities and relations randomly sampled? This approach may inadvertently query the LLM’s embedded knowledge (e.g., recognizing that alias A corresponds to B), which might not be present in the provided context.
3. Regarding sampling: How does the chosen sample of n=250 compare in ratio to the full knowledge graph? Additionally, how do we justify that this sample is unbiased, given that only the top 2000 nodes and edges are selected?

---

> ### Author Response · Authors · 2024-11-21
>
> We thank the reviewer for their time and constructive feedback. We address their concerns below. We hope our response mitigates their concerns and they consider increasing their support for our paper.
> > Deeper insights
>
> We would like to respectfully contradict the reviewer's view of our findings being predictable. The quantitative nature of our certificates enables deeper and novel insights, both model-specific and across models, some of which we mention next and also describe in the paper.
> 1. We find that the Phi3-3B model can do reasoning comparable to models with > 10B parameters, contradicting prior works [1,2].
> 2. We observe that quantization can deteriorate knowledge comprehension (e.g., 16% relative difference between performances of fp16 and 4-bit quantizations of Llama-3 8B in Table 1), contradicting prior works [3] which say that quantization preserves such capabilities.
> 3. We have added a new benchmarking baseline in Table 1 to compare with certification and highlight the latter's advantages. Details are in "new experiments" section of the general response.
>
> > More details of queries/context in main paper.
>
> Please check updated Section 4.
> > Parametric knowledge vs knowledge in context
>
> We agree with the reviewer that some queries can be answered by the LLM using its parametric knowledge and knowledge provided in the context may be redundant. However, even using parametric knowledge will require LLMs to comprehend the query and refer to relevant parametric knowledge to give the final correct answer. To the best of our knowledge, there is no way to definitely tell which knowledge source was used by the LLM. However, the knowledge provided in the context ensures that the LLM has sufficient information to answer the query correctly.
>
> > Random sampling of aliases and whether LLMs can link the aliases with the original entities.
>
> We randomly select entity and relation aliases from a set of aliases derived from Wikipedia pages. Random sampling ensures that we certify with respect to all the possible aliases, which can be realistically used by users in their prompts, with equal weightage. We agree that the LLMs might be unable to connect the aliases with the entities and relations based on just their embedded knowledge and hence provide information on that upfront in the context. We have updated the paper (Appendix B.3) with this detail.
>
> > Using LLM's generated conditional probability distribution
>
> We agree with the reviewer that evaluating the LLM responses with their generated probability distributions could be an interesting extension of our work. However, we chose to certify over the LLM's generated text for the following reasons. (1) Given that several SOTA LLMs showing good knowledge comprehension performance are closed-source, and thus do not provide the generated probabilities, the extension to using probability distributions can inhibit the applicability of the certification framework. (2) In line with most approaches for Question-Answering [4-6], we do greedy decoding for the LLM responses. Thus, the use of probability distributions would not add much to the analysis. Given that our method is based on several independent and identically distributed samples, which are sampled with replacement, we do not need to re-query the LLMs on the same prompt explicitly [7].
>
> > Clarification on 250 samples for certification
>
> We would like to clarify that *n=250 samples are used for 1 certification* for a property defined on a given subgraph of a knowledge graph. The certification guarantees are for prompts in the distribution defined over the subgraph, rather than the whole knowledge graph. Hence, the comparison between the samples in 1 certificate with the full knowledge graph is not well-defined, in our opinion. For our experiments, we extracted subgraphs from the Wikidata knowledge graph by performing bounded breadth-first searches with a maximum path length of $\rho$, starting from randomly selected pivot nodes. These pivots were drawn from two populations: the top 2000 nodes by out-degree in the global graph, and nodes whose subgraph within radius $\rho$ contains at least 2000 vertices (mentioned on line 301-305). Note, however, that these subgraphs were selected only for illustration purposes and our framework is not restricted to subgraphs having specific properties about them.
>
> ## References
> 1. Emergent abilities of large language models, Wei et al., 2022.
> 2. Tool learning with foundation models, Qin et al., 2024.
> 3. A comprehensive evaluation of quantization strategies for large language models, Jin et al., 2024.
> 4. Gemini: A Family of Highly Capable Multimodal Models, Gemini Team Google, 2024.
> 6. GSM-Symbolic: Understanding the Limitations of Mathematical Reasoning in Large Language Models, Mirzedeh et al., 2024.
> 7. TinyGSM: achieving > 80% on GSM8k with small language models, Liu et al., 2023.
> 7. Scalable Quantitative Verification For Deep Neural Networks, Baluta et al., 2021.

---

> > ### Comment · Reviewer_9uCh · 2024-11-21
> >
> > Thank you for your response and the additional experiments addressing the concerns. Below, I provide specific feedback on the revised work:
> >
> > **Parametric Knowledge vs. Knowledge in Context**
> >
> > While I appreciate the clarification regarding the difficulty in distinguishing between parametric knowledge and knowledge derived from the provided context, the reliance on parametric knowledge remains a significant factor. This is especially relevant considering large models have better parametric knowledge than smaller models, which influences all cross-model performance comparisons.
> >
> > To address this issue, I recommend masking all entities in the subgraph with random, unique strings (e.g., "Matthew Perry" → "UjKskuYd9v"). By doing so, the model would be exposed to entirely unseen entities, removing reliance on parametric knowledge and focusing solely on the comprehension of the given context. (a stricter variant could involve masking edge relations). This methodology could address the ambiguity and strengthen the claims about the LLM's context-based reasoning.
> >
> > ---
> >
> > **Bias in Subgraph Selection**
> >
> > While I understand the motivation behind selecting high-degree nodes or subgraphs with a minimum vertex count, this introduces a notable bias in the sampled subgraphs. Such nodes are more likely to represent popular entities or domains, which increases the likelihood of the LLM leveraging its embedded knowledge. I suggest justifying the influence of this selection bias in your paper and considering alternative sampling strategies in future work.
> >
> > ---
> >
> > **Overall Assessment**
> >
> > The additional experiments and clarifications improve the work, but there are still significant areas that require further refinement. While the updated submission demonstrates potential, in its current form, I believe it does not yet meet the standards for ICLR. I will maintain my rating for now, but I encourage the authors to continue developing these ideas.
> >
> > Thank you for your thoughtful response and for engaging with the feedback in detail.

---

> > > ### Author Response · Authors · 2024-11-24
> > >
> > > Thanks for the reviewer's constructive feedback on our rebuttal. We want to address the reviewer's concerns as follows.
> > >
> > > > Parametric versus in-context knowledge
> > >
> > > 1. **New experiment**. We conduct the suggested experiment to distinguish between the use of parametric knowledge versus in-context knowledge by the models, with a trivial extension of our framework, further demonstrating its generality. We provide the details next. We mask all entities in the Wikidata5m knowledge graph with random strings consisting of 10 characters comprising upper-case, lower-case letters and digits. We fix these strings for all entities in the knowledge graph, before any certification. We certify the 50 subgraphs with the masked entities, similar to our original method. As we have masked entities, we cannot use their original contexts, as they can reveal information about the original entity. Hence, we form a new context for each entity, consisting of descriptions of all relations it has with other entities. Such context consists of sentences like "{masked entity A} is related to {masked entity B} by relation {relation}", where relation {relation} exists between entities A and B. We prompt with a query with masked source node and give options consisting of masked entities, one of which is the (known) correct answer. We certify Gemini-1.5-Flash with QuaCer-C for the Wikidata5m knowledge graph with masked entities and obtain **[0.74,0.85]** as our average lower and upper bounds over 50 certificates, for the Vanilla setting (no information shuffling or distractors). These bounds are significantly higher than the corresponding average bounds in our original setting [0.46, 0.58] (from Table 1), suggesting the difficulty of unstructured and long context for the model. The improvement in the new results over the original ones suggests that the model may not have been using parametric knowledge to answer the queries, because if that was the case then the original bounds should have also been higher, irrespective of the challenges posed by the context. We can show more certification results in the new setting with entity masking, if the reviewer suggests. We believe that both entity and relation masking will make the task too unrealistic. However, we can show results for that too, if the reviewer suggests.
> > > 2. **Support for original approach**. Our original approach is in line with the design choices of the traditional open-book reading comprehension benchmarks [1-4]. Prior works have investigated the final question answering capabilities of the models, similar to us, irrespective of the use of parametric or in-context knowledge. The in-context knowledge is provided and models are encouraged to use the provided knowledge. Presence of in-context knowledge alleviates the need/absence of parametric knowledge, thus leveling the playing field for all models.
> > > 3. **Evidence of use of in-context knowledge in original approach**. The significant differences across the average certification bounds of different settings (Table 1) - Vanilla, Shuffle, and Shuffling with distractor, indicate that even in our original approach, the models are paying attention to the in-context knowledge and also getting distracted by additional information in the context (like distractor information).
> > >
> > > > Subgraph selection
> > >
> > > We understand the reviewer's concern and have hence updated the paper with appropriate justifications of our choice in the experiments (lines 302-305). Please note, however, that such choices do not undermine the efficacy of our framework, which is flexible to operate with any knowledge graphs.
> > >
> > > ## References
> > > 1. SQuAD: 100,000+ Questions for Machine Comprehension of Text, Rajpurkar et al., 2016.
> > > 2. RACE: Large-scale ReAding Comprehension Dataset From Examinations, Lai et al., 2017.
> > > 3. https://crfm.stanford.edu/helm/classic/latest/#/groups/natural_qa_openbook_longans
> > > 4. Can a Suit of Armor Conduct Electricity? A New Dataset for Open Book Question Answering, Mihaylov et al., 2018.

---

> > > > ### Comment · Reviewer_9uCh · 2024-12-02
> > > >
> > > > Thank you for the detailed response and masked entity experiment. Your justifications for subgraph selection and alignment with open-book benchmarks are convincing. Including additional masked entity results, if feasible, could further strengthen the work. I’ve increased my rating from 5 to 6.

---

> > > > > ### Author Response · Authors · 2024-12-02
> > > > >
> > > > > We are very grateful to Reviewer 9uCh for their appreciation of our response and experiments and for raising their score.

---

### Official Review · Reviewer_nuWN · 2024-11-05

**Soundness:** 4
**Presentation:** 3
**Contribution:** 2
**Rating:** 3
**Confidence:** 4

**Summary:**

This paper proposes to provide formal guarantees of model knowledge, as elicited by prompts derived from Wikidata. The object of the formal guarantee is the correctness of an answer to a question representing an arbitrary k-hop question stemming from some pivot node in the Wikidata knowledge graph. The means of formal guarantee is a binomial proportion confidence interval. To the best of my understanding, what this means is that the method guarantees model correctness with high confidence over a subgraph of Wikidata. The reason this requires a probabilistic guarantee, and cannot be done exhaustively, is that the combination of contexts for the questions, distractor text to provide alongside context, and few-shot examples for prompting the method creates a large prompt space that would be infeasible to exhaustively search. Experiments demonstrate that the authors can often bound model accuracy over a subgraph of Wikidata within about +/- .05 points.

**Strengths:**

- Important: I like the spirit of trying to give formal guarantees to model correctness for LLMs, which are difficult to handle analytically. The approach of using binomial proportion confidence intervals is a simple but appropriate one.
- Important: Experiments are carefully designed to demonstrate the main claims of the paper. A wide variety of models are tested.

**Weaknesses:**

- Important: While the main result in this paper is interesting, I also find it hard to say that it is especially impactful. The basic approach is to use a binomial proportion confidence interval to  estimate a model accuracy over a data distribution. The only way that this setting differs from any typical ML benchmark is that the authors define a data distribution over a subnetwork of Wikidata. As the authors note in L.242, longer paths in k-hop questions can result in somewhat surprising or meaningless queries. So I ask, what is really the point of certifying knowledge over such a subgraph? As in the qualitative example, we are not certifying knowledge about a movie. Rather, we are certifying knowledge about a movie, as well as a surprisingly diverse set of entities that are related to the movie. And, even if we were certifying knowledge about a movie, the next question is if the method in this paper merits publication if it primarily just makes use of an existing analytical binomial proportion confidence interval.
- Of some importance: While I believe the central point that we cannot exhaustively test deep learning models over input spaces is well-received, the paper has to introduce some complexity in order to make this difficulty appear in the first place in their setting. Specifically, aliases, contexts, distractors, and few-shot examples are randomly ordered in order to make the input space too large to exhaustively search. I believe it would also be possible to fix a basic set of instructions for strong models like Gemini and do these questions zero-shot. In that setting, there would not be a large combinatorial space to explore. Or, it might be more appropriate to generate model-based paraphrases of the input question, which may be more naturally representative of knowledge query diversity than the chosen approach.

**Questions:**

- L.65: strictly speaking, it’s not just the high number of parameters, right? Also nonlinearities?
- What makes R in L.274 a random variable? Is the any(.) operator effectively a uniform distribution? How is it defined? Later, when the paper says “we equally prioritize the different possible path lengths”, does this mean that the any(.) operator appropriately reflects this choice, or is there any bias in the estimator?
- Why use a multiple-choice format? Is the task too difficult otherwise?

---

> ### Author Response · Authors · 2024-11-21
>
> We thank the reviewer for their time and constructive feedback. We address their concerns below. We hope our response mitigates their concerns and they consider increasing their support for our paper.
> > Novelty and comparison with benchmarks
>
> Please refer to general response.
> > Certifying over subgraphs
>
> The idea behind certifying over subgraphs is not just to certify LLMs for answering queries related to the pivot node ("movie" from reviewer's example). It is to check whether the LLM can extract and reason over various pieces of information, starting at a pivot node to answer questions related to the pivot node. This is with various realistic corruptions of prompts, such as information shuffling, aliases, and distractor information. LLMs, which are extensively used in QA tasks, should, even under corruptions, be able to robustly provide correct answers. While we desire this accuracy over all possible QA tasks, we specify this as separate properties over local subgraphs, in line with majority works in neural network certification [1,2], to make the certification tractable. Note that, as we illustrate that longer paths can become meaningless, we restrict the maximum path lengths starting from pivot node in subgraphs over which we certify, to 5 (mentioned in Section 4.1). We observe that queries over paths longer than 5 become quite distinct from the pivot node. Our framework is not restricted to just subgraphs, however, and we show additional certification results for specifications over paths having same relations but varying entities, in "new experiments" in general response.
> > Intractable input space
>
> Aliases, unstructured context, and distractors are realistic corruptions that can occur in practical user prompts and standard datasets [3,4] also contain instances of these. This necessitates accounting for random variations of aliases, information ordering, and distractor texts when certifying for knowledge comprehension. Restricting to specific prompt structures will give us limited understanding of knowledge comprehension by models (e.g., Gemini), and guarantees over the small input spaces will not be about practical user prompts. Just to clarify, we do not vary few shot examples across prompts.
> > Model-based paraphrases for certification
>
> Model-specific paraphrasing could be an alternative to get knowledge comprehension prompts for LLMs. As our specification (Algorithm 1) is agnostic to distributions D over aliases, model-specific distributions are special cases that can be used to certify.  That may also yield an intractable input space for realistic D. We do not make D model-specific to compare different models for knowledge comprehension on a common standard for fair evaluation. Moreover, as common users are not expected to deliberately apply model-specific corruptions to their prompts but rather introduce random corruptions inadvertently, we use the same input prompt spaces for all models.
> > Clarification of $\mathcal{R}$
>
> R is a function of randomly sampled prompt P (Algorithm 1, lines 1-5), and hence a random variable. It denotes whether LLM L can output any alias of the tail node of path $\Pi$ underlying P, when prompted with P. any(.) is a deterministic primitive of our specification language. We have updated our usage of any(.) in the specification (Algorithm 1) to be more understandable. It denotes that L's output for P matches any alias of the tail node $\Pi[-1]$ of $\Pi$, as all the aliases are correct answers. We have clarified any(.) in line 234 as well. We equally prioritize different path lengths when sampling P. any(.) simply checks whether L's answer for given P matches with the correct answer for corresponding path $\Pi$. As any(.) permits L to give any correct alias as answer, we do not see any bias in the certifier due to it.
> > Multiple-choice format?
>
> Our prompts contain multiple-choice questions (MCQs) similar to prior works on question-answering [5,6], which consider MCQ QA as an important task. The main challenge of free-form responses is accurate evaluation. Specifically, we need to check whether response mentions any of 100s of possible aliases of the correct answer as final answer. We observe several false evaluations, even by LLM-as-a-judge, for free-form responses. Hence, we evaluate with MCQ prompts. However, this is just an implementation detail and our theoretical framework generalizes to free-form responses too.
> > Including non-linearities
>
> We agree with the reviewer on the complexity added by non-linearities for LLM certification and have added this in line 64.
> ## References
> 1. Formal Specification for Deep Neural Networks
> 2. Fast and Precise Certification of Transformers
> 3. KEPLER: A Unified Model for Knowledge Embedding and Pre-trained Language Representation
> 4. Large language models can be easily distracted by irrelevant context
> 5. Measuring massive multitask language understanding
> 6. Think you have Solved Question Answering? Try ARC, the AI2 Reasoning Challenge

---

### Author Response · Authors · 2024-11-21
**General response (2) - Novelty**

> [nuWN,w1nm] Comparison with benchmarks

Following are the main points of difference of our framework over traditional KBQA benchmarks [4-6].
1. Avoiding test set leakage: Unlike benchmarking with standard KBQA datasets, the distribution-driven analysis of QuaCer-C provides more consistent and reliable assessment of knowledge comprehension. This is because, analysis with distributions avoids test set leakage, where models are trained on the test set. Prior works [1-3] have shown inconsistent benchmark analyses, which could be due to test set leakage. Distributions, however, are not fixed datasets that models can memorize during training. We sample from them by running probabilistic programs (Algorithm 1). Hence, if an LLM performs well in our setting, then it is less likely due to memorization.
3. Generalizability: QuaCer-C's high confidence bounds generalize over prohibitively large distributions. Conventional benchmarks can not scale to such distributions, due to their enumerative analysis.
4. Holistic analysis with varying challenges: Typical KBQA datasets capture only limited kinds of potentially adversarial corruptions of prompts, such as using different aliases [4], distractors [5], and information ordering [6]. Extending them to include more challenges requires significant manual effort. However, our distributions are generalizations of various kinds of corruptions and enable a holistic study of knowledge comprehension.
5. Insights on worst and best-case performance: We provide a new baseline on benchmarking with a static dataset, in the "new experiments" section of the general response. We see that the baseline is an optimistic view of knowledge comprehension in LLMs, missing out on several failure cases of the models. Certification with QuaCer-C can, however, indicate the worst and best-case performance of the models with certification bounds, constituting a robust assessment. The average certification bounds can be used as measures of robust knowledge comprehension alongside benchmarking results, like prior works on certifying neural networks [7,8].

>[nuWN] Novelty

As reviewer nuWN points out, we design novel distributions and their samplers over related problems to specify correct knowledge comprehension by target LLMs. As we believe and prior works [9,10] discuss, designing sampleable input distributions is important to establish desirable properties over given ML models and crucial for probabilistic certification [11,12]. For specialized properties, like correct knowledge comprehension, we need specific prompt distributions, which can capture natural, challenging prompts requiring the property and from which we can efficiently obtain independent and identically distributed (iid) samples. Our distributions are the first prompt distributions for knowledge comprehension, to the best of our knowledge. They capture realistic but challenging prompts with long, unstructured context by incorporating entity aliases (from Wikipedia pages), information shuffling, and distractor information. Moreover, their sampler (Algorithm 1, lines 1-5), can efficiently generate iid samples.

We do not claim novelty in the statistical estimation method used, but rather in enabling its use for certifying knowledge comprehension with novel distributions. Binomial proportion confidence intervals require iid samples from distributions over which estimation is done. While we find them suitable for certifying LLMs (nuWN also agrees to their suitability), leveraging them directly without distributions of prompts for knowledge comprehension is not possible. This is because standard datasets do not guarantee that their elements are sampled iid. We strongly believe that identifying and enabling an existing statistical estimation algorithm to provide the first formal guarantees for the important property of knowledge comprehension in LLMs can be a valuable contribution towards trustworthy LLMs. Prior works, such as [7,11] which have also used existing statistical methods for trustworthy AI have been well-received by ICML and ICLR and have been very impactful.
## References
1. Larger language models do in-context learning differently
2. In-context learning and induction heads
3. Why larger language models do in-context learning differently
4. Large language models can be easily distracted by irrelevant context
5. Constructing Datasets for Multi-hop Reading Comprehension Across Documents
6. KEPLER: A Unified Model for Knowledge Embedding and Pre-trained Language Representation
7. Certified Adversarial Robustness via Randomized Smoothing
8. Property-Driven Evaluation of RL-Controllers in Self-Driving Datacenters
9. Adversarial Distributional Training for Robust Deep Learning
9. NATTACK: Learning the Distributions of Adversarial Examples for an Improved Black-Box Attack on Deep Neural Networks
9. A Statistical Approach to Assessing Neural Network Robustness
9. Probabilistic Verification of Fairness Properties via Concentration

---

### Author Response · Authors · 2024-11-21
**General response**

Dear Area Chair and Reviewers,

We thank the reviewers for their time and positive feedback. We are encouraged by the reviewers finding our work to be well-formulated and extensively evaluated.

For the Area Chair: As per the reviewers' suggestions, we have included additional experiments (shown below) and have updated our paper with the details. We are happy to conduct more experiments and provide further clarifications, if needed. The revised paper now contains the Appendix attached to the main paper (instead of supplementary material). The text added to the revised paper is highlighted in red color.

## Updates to paper
1. New baseline results added to Section 4 and Table 1.
2. More details on query and context construction added to Section 4 of main paper from the Appendix.
3. Appendix A.4 added with new chain-of-thought experiments.

## New experiments
> [9uCh] Benchmarking baseline

We include a new benchmarking baseline in Table 1 (Section 4) of the revised paper. The baseline results give the accuracy of LLMs on a static dataset formed with randomly-selected paths in the subgraphs of Wikidata5m which we also use in our certification results in Table 1. The baseline setup is described further in Section 4, Lines 338-342 of the revised paper. Comparison with benchmarking baseline reveals important discrepancies. Firstly, the baseline shows an optimistic view of LLM performance for knowledge comprehension, with high numbers. It fails to capture many vulnerabilities of LLMs due to evaluation on a fixed dataset with limited challenges. Certification, however, explores various vulnerabilities of the models by evaluating with multiple prompts with varying difficulties. Moreover, baseline testing shows Mistral-7B outperforming Phi3-3B, as well as scores that exceed the certification upper bounds. Additionally, Phi3-14B's 8-bit quantized version shows large performance drops when compared with the fp16 model in baseline testing. These results suggest that standard benchmarking methods may present only a limited picture of the performance of LLMs for knowledge comprehension, sensitive to the particular details of constructing the prompts and the static datasets.

> [mSM5] Additional models with varying size from same family

We certify Llama-3.2-Instruct 1B, 3B, and 11B models to substantiate our claim on the improvement in knowledge comprehension with model size, within the same model family. Unfortunately, we do not have the resources to certify much larger open-source models. The performance show clear improvement from the 1B to 3B to 11B models with average certification bounds - (0.24, 0.35); (0.30, 0.41); (0.34, 0.46) respectively, demonstrating that increasing model size leads to better performance.

> [mSM5] Use Chain of Thought (COT)

We apply COT to Phi-3 (3B) in the vanilla (no information shuffling/distractors) setting. We provide details of our experimental setup and results in Appendix A.4. As anticipated, performance improved, yielding a new range of (0.44, 0.56) - a *10% increase* in both average lower and upper bounds. This highlights the broader applicability of our framework, which is compatible with various prompting techniques and models. While we acknowledge the potential benefits of COT, earlier experiments were limited due to the significantly increased computational cost of COT (generating 5-8 times more tokens), particularly with closed-source models as output tokens are expensive.

> [mSM5] Certification of specifications with constant path relations

The suggested specification can be certified with a trivial extension of our framework. We show 2 example certificates next. We certify Phi-3 (3B) over queries formed from paths of the kind - "...$\rightarrow$(appeared in movie)$\rightarrow$(directed by)$\rightarrow$?" (suggested by mSM5) and "...(host country)$\rightarrow$(flag description)$\rightarrow$?" (very common path in Wikidata5m). Specifically, we (uniformly) randomly select paths having the above structure but varying entities that are related by the aforementioned relations. We form queries from these paths by selecting (uniformly) random aliases of the path entities. We generate the Clopper-Pearson confidence intervals with $250$ prompt samples from the distribution over such paths and obtain the certification bounds - (0.74, 0.84) and (0.68, 0.80), respectively. We hypothesize that the bound values are high because the paths are simple and common ones in the knowledge graph, involving just $2$ reasoning steps. Paths with higher number of reasoning steps, agnostic of the individual entities, are less common in the knowledge graph. We are happy to provide more example certificates of similar properties, if the reviewer suggests. However we are unable to put these certification results in the paper, as it would not be compatible with the current writing of the theoretical sections.

---

> ### Comment · Reviewer_nuWN · 2024-11-25
>
> > These results suggest that standard benchmarking methods may present only a limited picture of the performance of LLMs for knowledge comprehension
>
> It sounds like the main claim here is that the proposed random prompts are _more accurate measurements_ than the baseline benchmarks, due to e.g. having prompts of varying difficulty. This feels a little orthogonal to the point of knowledge certification. While it is good to have benchmarks vary prompts by varying entity aliases, including distractor information, and shuffling information order, this has more to do with making benchmarks representative of settings that we care about, and less to do with certifying performance.
>
> > Certification of specifications with constant path relations
>
> I think this addition to the paper is nice. It could help show that models understand a certain kind of (multi-hop) relationship between entities in the world.
>
> > If an LLM performs well in our setting, then it is less likely due to memorization.
>
> Actually, are answers always clearly derivable from the contexts? I agree with the idea that varying the surface form realization of some underlying question can help mitigate test set leakage for LLMs. Just to keep the terminology clear, doing well on wikidata-based tasks should be pretty correlated with a model having "memorized" Wikipedia, in the sense that the model has memorized factual information from Wikipedia, even when it's not the case that the exact test-set prompts were seen in the training data.
>
> > main points of difference of our framework over traditional KBQA benchmarks
>
> Overall, I think these points do not fully convince me that this certification framework is a better way of benchmarking model knowledge, reading comprehension, or k-hop QA ability. I like the certification idea at its core, but I feel like this point might be better illustrated by showing something like: (1) researcher 1 benchmarks an LLM on some typical knowledge benchmarks, and (2) researcher 2 tries to replicate the benchmark results with prompts that differ in reasonable ways, using some of the transformations described in this paper. A certificate from researcher 1 could help researcher 2 understand whether their results are surprising or not, if they highly over/underperform numbers from researcher 1. I understand this is basically a different paper from the one here, focused on off-the-shelf knowledge benchmarks and not k-hop QA based on wikidata. Since the authors bring up the importance of the "corruptions" multiple times in the rebuttal, while maintaining that the certificates are useful but not strictly novel, I am trying to imagine an application of both of these directions that could produce a more compelling or widely useful final result than certifying k-hop performance over wikidata.

---

> > ### Author Response · Authors · 2024-11-27
> >
> > We thank reviewer nuWN for their insightful comments on our general response. We would, however, like to clarify our position on their comments as follows.
> > > Comparing certification and benchmarking wrt prompting strategies
> >
> > We do not necessarily claim that certification is more accurate than benchmarking. It provides guarantees on LLM performance with respect to input distributions. Hence it can complement benchmarking for detailed insights into knowledge comprehension capabilities of LLMs for distributions on which benchmarking can not scale, like prior certification works for traditional ML [1-3].
> > Our framework QuaCer-C is general, not restricted to specific prompt distributions with varying aliases, distractors, or shuffling. Distributions over knowledge comprehension prompts are separate contributions from certification algorithm.
> > **Comparing datasets with distributions**. Our distributions succinctly represent large number (~$10^{16}$) of prompts, which can not be scalably enumerated in benchmarks. Moreover, variations of aliases, distractors, etc., are natural perturbations, which can occur in realistic user prompts. Prior works such as [3] have also used programmatic representations of natural language perturbations to certify LSTM classifiers. Hence, we believe that certifying over natural prompt perturbations is important to robustly assess knowledge comprehension capabilities, amidst varying prompt complexity.
> > > Are answers in contexts
> >
> > Yes, correct responses can always be derived from context in prompts. Details of context construction are in Appendix B.4.
> > > Clarification on test set leakage and memorization
> >
> > We believe that in-general certification with prohibitively large distributions (like our proposed distributions) does not suffer from test set leakage or memorization issues, as training on all underlying prompts may not be possible.
> > For our distributions specifically, we resonate with the reviewer that variations of prompts due to long, unstructured context, aliases, distractors, and information shuffling, change the structure of prompts sufficiently to avoid test set leakage. Overall LLM responses can be derived only after accurately denoising all challenges in prompts, which may be non-trivial. Moreover, to study whether models use in-context or memorized (parametric) knowledge, we conduct a new experiment (suggested by Reviewer 9uCh) described below.
> > > Utility of certification over benchmarking.
> >
> > We thank the reviewer for their insights on utilizing certificates to reconcile variable benchmarking results. We will definitely look into this in future work.
> > We believe that certification is important by itself, irrespective of its use with other evaluation methods.
> > 1. **General utility of certificates**. Unlike benchmarking, certification is a reliable evaluation method, which also gives uncertainty of knowledge comprehension assessment (with bounds) and statistical guarantees that generalize over (large) given input distributions. Recent work [4] from Anthropic suggests shift of trends in industry towards statistical methods, which is the core of QuaCer-C.
> > 2. **Utility of certificates over distributions of k-hop QA over Wikidata**. Our distributions are defined over Wikidata as it is a popular, open-source, large knowledge graph. However, our framework is not restricted to Wikidata alone and can be extended to other knowledge graphs. Prior benchmarking studies [5,6] have also investigated the k-hop QA capabilities of language models as it is traditionally considered an important capability. Hence, we believe that certifying over distributions of k-hop QA can provide new insights into the k-hop QA problem, which were not available with benchmarking alone. Corruptions, such as aliases, distractors, shuffling, are natural prompt perturbations that have been studied in prior works [7-9] with limited applicability. We develop general QA distributions including them, to make the certification more comprehensive and practically useful.
> > ## References
> > 1. Certified Adversarial Robustness via Randomized Smoothing
> > 2. Property-Driven Evaluation of RL-Controllers in Self-Driving Datacenters
> > 3. Certified Robustness to Programmable Transformations in LSTMs
> > 4. Adding Error Bars to Evals: A Statistical Approach to Language Model Evaluations
> > 5. Constructing Datasets for Multi-hop Reading Comprehension Across Documents
> > 6. HOTPOTQA: A Dataset for Diverse, Explainable Multi-hop Question Answering
> > 7. Large language models can be easily distracted by irrelevant context
> > 8. Constructing Datasets for Multi-hop Reading Comprehension Across Documents
> > 9. KEPLER: A Unified Model for Knowledge Embedding and Pre-trained Language Representation

---

> ### Author Response · Authors · 2024-11-27
> **New experiment on parametric vs in-context knowledge**
>
> We conduct the following experiment to distinguish between the use of parametric knowledge versus in-context knowledge by the models. We mask all entities in the Wikidata5m knowledge graph with random strings consisting of 10 characters, with a combination of upper-case, lower-case letters and digits. We certify the 50 subgraphs with the masked entities, similar to our original method. As we have masked entities, we cannot use their original contexts, as that can reveal information about the original entity. Hence, we form a new context for each entity, consisting of descriptions of all relations it has with other entities. Such context is structured as "{masked entity A} is related to {masked entity B} by relation {relation}", where relation {relation} exists between entities A and B. We asked a query with masked source node from the model and give it options consisting of masked entities, one of which is the correct answer. We certify Gemini-1.5-Flash model with QuaCer-C for the Wikidata5m knowledge graph with masked entities and obtain **[0.74,0.85]** as our average lower and upper bounds over 50 certificates, for the Vanilla setting (no information shuffling or distractors). These bounds are significantly higher than the corresponding average bounds in our original setting [0.46, 0.58] (from Table 1), suggesting the difficulty of unstructured and long context for the model. The improvement in the new results over the original ones suggests that the model may not have been using parametric knowledge to answer the queries, because if that was the case then the original bounds should have been higher, irrespective of the challenges posed by the context. We can show more certification results in the new setting with entity masking, if the reviewers suggest.

---

### Meta-Review · Area_Chair_52hJ · 2024-12-21

**Metareview:**

The paper introduces QuaCer-C, a framework for certifying knowledge comprehension in LLMs using knowledge graphs. While the approach attempts to provide statistical confidence in model responses to knowledge comprehension prompts, reviewers highlighted several weaknesses. Key concerns included the limited practical utility of the certification process due to its dependence on structured datasets like knowledge graphs, lack of novelty compared to existing KBQA methods, and unclear theoretical justifications. I recommend rejection.

**Additional Comments On Reviewer Discussion:**

During the rebuttal, the authors clarified the novelty of QuaCer-C, which partially addressed the reviewers' concerns. However, the responses failed to fully convince certain reviewers, such as nuWN (an expert reviewer).

---

### Decision · Program_Chairs · 2025-01-22

Reject